# Satb2 determines miRNA expression and long-term memory in the adult central nervous system

Clemens Jaitner[1†], Chethan Reddy[1†], Andreas Abentung[1†], Nigel Whittle[2], Dietmar Rieder[3], Andrea Delekate[4], Martin Korte[4,5], Gaurav Jain[6,7], Andre Fischer[6,8], Farahnaz Sananbenesi[7], Isabella Cera[1], Nicolas Singewald[2], Georg Dechant[1*‡], Galina Apostolova[1*‡]

[1]Institute for Neuroscience, Medical University of Innsbruck, Innsbruck, Austria; [2]Department of Pharmacology and Toxicology, University of Innsbruck, Innsbruck, Austria; [3]Division of Bioinformatics, Biocenter, Medical University of Innsbruck, Innsbruck, Austria; [4]Zoological Institute, Technical University Braunschweig, Braunschweig, Germany; [5]AG Neuroinflammation and Neurodegeneration (NIND), Braunschweig, Germany; [6]Research Group for Epigenetics in Neurodegenerative Diseases, German Center for Neurodegenerative Diseases, Göttingen, Germany; [7]Research Group for Complex Neurodegenerative Disorders, German Center for Neurodegenerative Diseases, Göttingen, Germany; [8]Department of Psychiatry and Psychotherapy, University Medical Center, German Center for Neurodegenerative Diseases, Göttingen, Germany

**\*For correspondence:** georg.dechant@i-med.ac.at (GD); galina.apostolova@i-med.ac.at (GA)

[†]These authors contributed equally to this work
[‡]These authors also contributed equally to this work

**Competing interests:** The authors declare that no competing interests exist.

**Abstract** *SATB2* is a risk locus for schizophrenia and encodes a DNA-binding protein that regulates higher-order chromatin configuration. In the adult brain Satb2 is almost exclusively expressed in pyramidal neurons of two brain regions important for memory formation, the cerebral cortex and the CA1-hippocampal field. Here we show that Satb2 is required for key hippocampal functions since deletion of Satb2 from the adult mouse forebrain prevents the stabilization of synaptic long-term potentiation and markedly impairs long-term fear and object discrimination memory. At the molecular level, we find that synaptic activity and BDNF up-regulate Satb2, which itself binds to the promoters of coding and non-coding genes. Satb2 controls the hippocampal levels of a large cohort of miRNAs, many of which are implicated in synaptic plasticity and memory formation. Together, our findings demonstrate that Satb2 is critically involved in long-term plasticity processes in the adult forebrain that underlie the consolidation and stabilization of context-linked memory.

## Introduction

Satb2 is a transcriptional regulator that binds to matrix attachment regions in the DNA and recruits chromatin-modifying complexes at the anchorage sites (*Baranek et al., 2012*; *Britanova et al., 2005*; *Gyorgy et al., 2008*; *Szemes et al., 2006*). Furthermore, similarly to its homologue Satb1 (*Wang et al., 2014*, *2012*), Satb2 modifies higher-order chromatin structure by mediating the formation of intra-chromosomal DNA loops (*Zhou et al., 2012*).

Recent genome-wide association studies of schizophrenia have identified *SATB2* as a genetic risk locus (*Schizophrenia Working Group of the Psychiatric Genomics Consortium, 2014*). Moreover, patients with mutations or deletions within the *SATB2* locus, a condition referred to as 'SATB2-

associated syndrome (SAS)', exhibit severe learning difficulties and profound mental retardation, providing further indication for a potential role of SATB2 in higher brain function (*Liedén et al., 2014*; *Zarate et al., 2015*; *Zarate and Fish, 2016*; *Marshall et al., 2008*). So far the neuropsychiatric symptoms of SAS have been discussed in the context of the established role of Satb2 during embryonic development of the cerebral cortex. In the embryonic cortex Satb2 is restricted to upper layer neurons where it inhibits the corticospinal motor neuron fate and promotes callosal neuron identity (*Alcamo et al., 2008*; *Britanova et al., 2008*; *Leone et al., 2015*; *Srinivasan et al., 2012*; *Srivatsa et al., 2014*). Thus, deficits in cortico-cortical connections could account for the reported neurological defects in SAS patients. However, patients with *SATB2* haploinsufficiency have no apparent corpus callosum abnormalities (*Lee et al., 2015*; *Rosenfeld et al., 2009*). In heterozygous Satb2 knockout mice, resembling the genetic condition of SAS patients, the corpus callosum is also intact (*Alcamo et al., 2008*). This suggests a function of Satb2 in adult brain independent from its developmental role. The function of Satb2 in the adult central nervous system (CNS) is completely unknown since germ-line Satb2-deficient mice die perinatally (*Dobreva et al., 2006*). In contrast to the layer-specific embryonic expression, adult CNS Satb2 is expressed in pyramidal neurons of all layers of the cerebral cortex and in the hippocampal CA1 area (*Huang et al., 2013*). As both brain regions are tightly linked to cognition, Satb2 is well-positioned to regulate cognitive processes.

In this study, we investigated the role of Satb2 in the mature mouse brain by selectively deleting *Satb2* from forebrain excitatory neurons after the third postnatal week. Our results demonstrate deficient long-term potentiation (LTP) and long-term memory in Satb2 conditional mutants. At a mechanistic level, we establish Satb2 as a nuclear component of two main pathways implicated not only in cognition but also in schizophrenia pathophysiology, i.e. BDNF signaling and miRNA-mediated post-transcriptional regulation of gene expression.

## Results

### Satb2 is necessary for long-term memory formation and hippocampal late-LTP

Given the highly specific expression pattern of Satb2 in the adult brain (*Figure 1A*) as well as the severe learning disabilities and mental retardation observed in SAS patients, we hypothesized that Satb2 is critical for learning and memory. To circumvent the perinatal and early postnatal lethality of the existing constitutive and conditional Satb2 mutants (*Dobreva et al., 2003*; *Srinivasan et al., 2012*) and to be able to perform behavioral experiments, we generated a novel conditional Satb2 mutant line by crossing mice bearing a floxed allele of *Satb2* (*Satb2*flox/flox) with mice that express *Cre* recombinase under the *Camk2a* promoter (*Minichiello et al., 1999*). The expression of the *Camk2a-Cre* transgene allowed for a forebrain-specific deletion of Satb2 from the third postnatal week on, thus bypassing the confounding effects of early Satb2 inactivation on the formation of cortical neuronal circuits (*Alcamo et al., 2008*; *Britanova et al., 2008*; *Harb et al., 2016*; *Leone et al., 2015*; *Srinivasan et al., 2012*; *Srivatsa et al., 2014*). The absence of Satb2 protein in the cortex and hippocampus of adult but not juvenile *Satb2*flox/flox*::Camk2a-Cre* mice (Satb2 cKO) was confirmed by immunoblotting (*Figure 1B*) and immunohistochemistry (*Figure 1C*, *Figure 1—figure supplement 1*). Satb2 cKO mice were viable, fertile, and reached the same age and body weights as their littermate controls (*Figure 1—figure supplement 2A*). Gross morphological examination revealed no abnormalities in the brain of Satb2 cKO mutants (*Figure 1—figure supplement 2B*). Corpus callosum and the cellular layers of the neocortex and hippocampus appeared intact (*Figure 1D*). Immunoreactivity for the CA1-specific marker Wfs1 (*Figure 1E*) and the cortical layer markers Cux1, Ctip2, Tbr1 (*Figure 1—figure supplement 2C*) was undistinguishable from control mice, suggesting normal cortical and hippocampal morphology in Satb2 conditional mutants.

To test whether Satb2 is required for learning and memory we used contextual fear conditioning, a hippocampus-dependent paradigm of associative learning. Satb2 cKO mice showed a normal response to electric foot shock exposure (*Figure 2—figure supplement 1*) and normal fear acquisition during fear conditioning (*Figure 2A*). Short-term memory for contextual fear, measured 1 hr following training, was also not affected in Satb2 cKO mice. However, Satb2 cKO mice exhibited a significant decrease in freezing when compared to control littermates 24 hr after training (*Figure 2A*), indicating a specific deficit in the consolidation of associative memory. Next, we

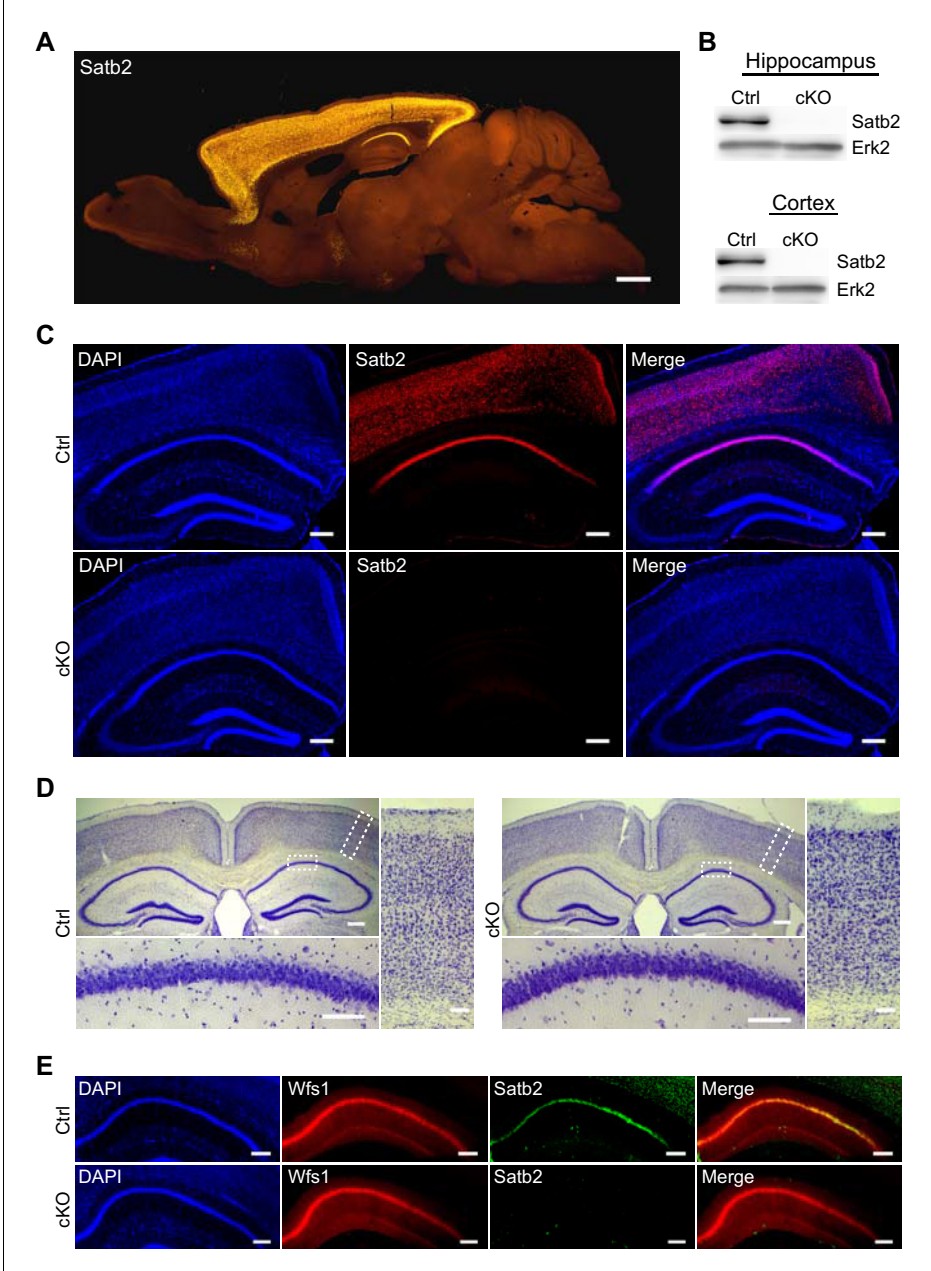

**Figure 1.** Characterization of Satb2 conditional mutants. (**A**) Satb2 is mainly expressed in the adult forebrain. Immunostaining for Satb2 of sagittal brain sections from adult mice. Scale bar: 1000 μm. (**B**) Immunobloting analysis of the Satb2 protein level in cortical or hippocampal lysates from adult Satb2 cKO mice (cKO) or *Satb2*[flox/flox] mice (Ctrl). Erk2 was used as loading control. Representative images are shown. (**C**) Satb2 immunostaining of coronal brain sections from 3-month old Satb2 cKO mice or *Satb2*[flox/flox] littermate controls. Nuclei were counterstained with DAPI. Representative images are shown. Scale bar: 200 μm. (**D**) Nissl-stained coronal brain sections from 3-month old Satb2 cKO mice and littermate controls, demonstrating normal gross brain morphology of Satb2 cKO animals. Scale bar: 200 μm. High magnification views of boxed areas reveal the normal cyto-architecture of the cortex and hippocampus of Satb2 cKO mice. Representative images are shown. Scale bar: 50 μm. (**E**) Immunohistochemical labeling against the CA1 specific marker Wfs1 in hippocampus of Satb2 cKO mice and littermate controls reveals normally developed hippocampal CA1 area in Satb2 mutants. Nuclei were counterstained with DAPI. Representative images are shown. Scale bar: 150 μm.

The following figure supplements are available for figure 1:

*Figure 1 continued on next page*

*Figure 1 continued*

**Figure supplement 1.** Satb2 is expressed in the cortex and hippocampus of both Satb2 conditional mutants and littermate controls at postnatal day 15.

**Figure supplement 2.** Postnatal Satb2 deletion does not cause alterations in body weight, gross brain morphology and cortical layer-specific marker expression.

subjected Satb2 conditional mutants to the object location memory (OLM) and novel object recognition memory (ORM) tasks. Again, Satb2 cKO mice demonstrated normal short-term (1 hr) memory but significant deficits in long-term (24 hr) OLM (*Figure 2B*) and ORM (*Figure 2C*), providing evidence for requirement of Satb2 for long-term object discrimination/placement memories.

Next, we investigated the effect of loss of Satb2 on long-term potentiation (LTP), an electrophysiological correlate of memory formation (*Mayford et al., 2012*). To this aim, we prepared acute slices from Satb2 cKO mice and littermate control animals and tested LTP at hippocampal Schaffer collateral-CA1 synapses. Field-excitatory post-synaptic potential recordings from the apical dendritic layer of the CA1 region showed that the early phase of LTP (up to 40 min post-theta burst stimulation) did not differ from control values; however late-LTP (45–180 min post-theta burst stimulation) was significantly attenuated in Satb2 cKO mice (*Figure 3A*). The slopes of the input-output curves (*Figure 3B*) and the paired-pulse facilitation ratios across different inter-pulse intervals (*Figure 3C*) did not differ between mutant and control mice indicating normal basal synaptic transmission and presynaptic function in Satb2 cKO mice. Hence, Satb2 is not required for short-term plasticity at CA3–CA1 synapses but is essential for late-LTP maintenance.

## BDNF and synaptic activity up-regulate Satb2 via the ERK1/2 pathway

Given our findings of impaired hippocampal late-LTP in Satb2 conditional mutants, we examined whether neuronal activity or the neurotrophin BDNF, a mediator of structural and functional plasticity at synapses (*Zagrebelsky and Korte, 2014*), regulate Satb2 in primary hippocampal neurons.

We treated hippocampal cultures with bicuculline (Bic) and 4-aminopyridine (4AP), a combination that causes robust action potential (AP) bursting (*Hardingham et al., 2002*). Bic/4AP application for 24 hr resulted in a strong up-regulation of both Satb2 mRNA (2.5-fold change, $n = 7$, Student's *t* test, $p = 0.003$) and protein levels (6.3-fold change, ANOVA, Ctrl vs. Bic/4AP, $p = 0.002$, *Figure 4A*). Next, we applied MK-801, a NMDA receptor antagonist, or nimodipine, an L-type VGCC blocker, together with Bic/4AP for 24 hr to determine which source of calcium (*Bengtson et al., 2013*) is required for Bic/4AP-triggered Satb2 induction. Nimodipine, but not MK-801, blocked the increase of Satb2 following synaptic activity indicating that Satb2 induction after synaptic stimulation depends on calcium influx through L-type VGCC and not through NMDA receptors (*Figure 4A*). Also BDNF and NT4, which both bind to the tyrosine kinase TrkB receptor, caused a significant increase in Satb2 mRNA (1.8-fold change, $n = 7$, Student's *t* test, $p = 0.003$) and protein (3-fold change, ANOVA, Ctrl vs. BDNF, $p = 0.0004$; Ctrl vs. NT4, $p = 0.0002$) levels 24 hr after treatment (*Figure 4B*).

Since BDNF expression and secretion in the CNS are controlled by neuronal activity we reasoned that the induction of Satb2 following AP bursting might be due to enhanced Bdnf transcription, translation and/or BDNF release (*Hardingham et al., 2002*; *Kuczewski et al., 2009*; *Tao et al., 1998*). To further investigate this possibility we inhibited Trk signaling with K252a during Bic/4AP stimulation. As a control experiment, we pharmacologically blocked AP bursting with the sodium channel blocker tetrodotoxin (TTX) during BDNF treatment. We found that silencing the neuronal activity with TTX did not affect BDNF-induced Satb2 expression, even though it abolished the synaptic activity-driven Satb2 induction (*Figure 4C*). In contrast, synaptic activity-triggered increase of Satb2 was blocked by Trk antagonism (*Figure 4D*), suggesting that synaptic activity-triggered Satb2 induction is indeed mediated via BDNF/TrkB signaling. De novo gene transcription is necessary for this process since actinomycin D, an inhibitor of gene transcription, blocked BDNF-driven Satb2 up-regulation (*Figure 4—figure supplement 1*).

To determine if MEK1/2 - ERK1/2 pathway (*Minichiello, 2009*) contributes to BDNF-triggered Satb2 induction we applied the MEK1/2 inhibitor UO126 1 hr prior to BDNF stimulation. UO126

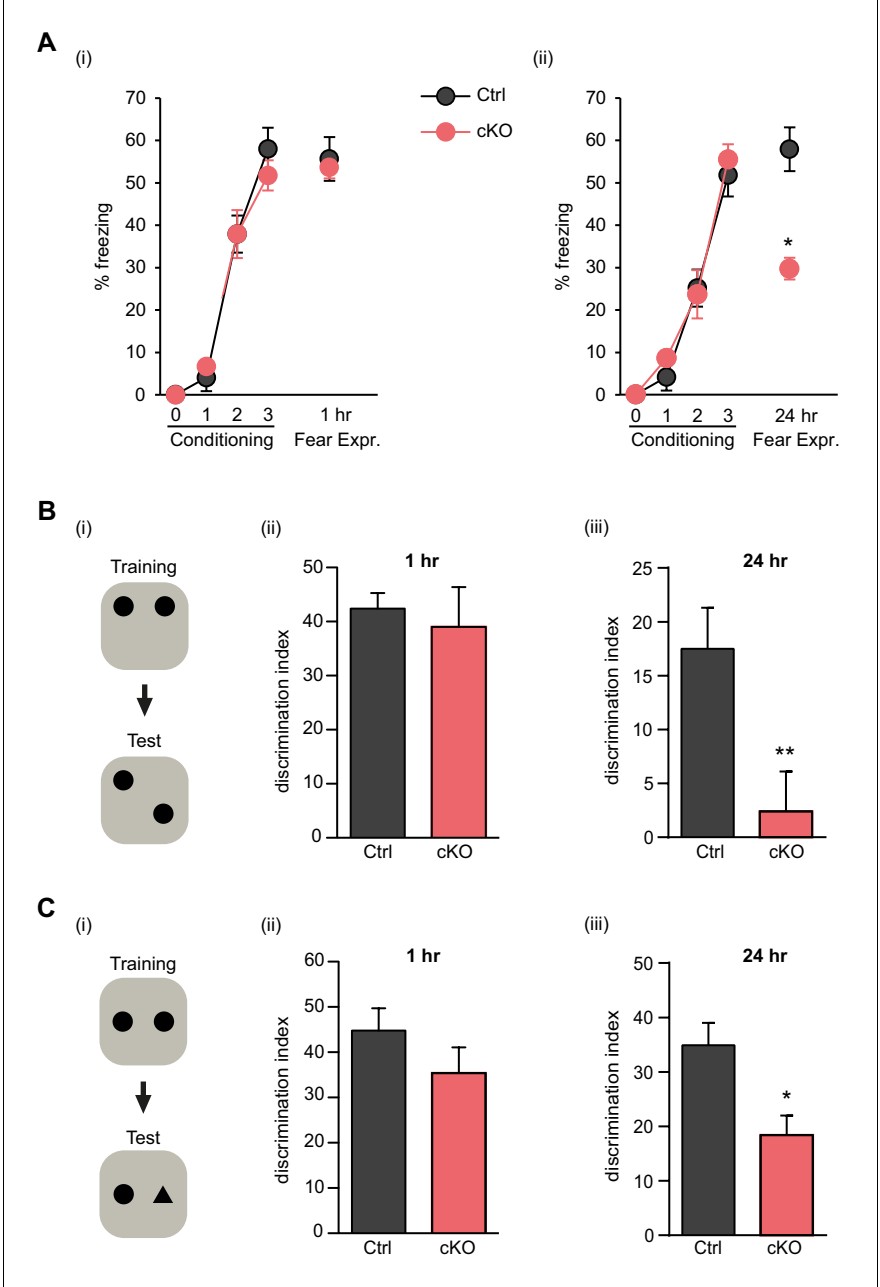

**Figure 2.** Satb2 is required for long-term memory formation. (**A**) In a contextual fear conditioning paradigm, Satb2 cKO mice (cKO), showed (**i**) similar levels of freezing to *Satb2*[flox/flox] mice (Ctrl) during the fear-acquisition phase (cKO, *n* = 7; Ctrl, *n* = 6; repeated measures ANOVA, $F_{3,33} = 0.76$, p = 0.52) and at the 1 hr fear expression test (Student's *t* test, $t_{11} = 0.19$, p = 0.86) but (**ii**) froze significantly less than their littermate controls at the 24 hr fear expression test (cKO, *n* = 8; Ctrl, *n* = 8; repeated measures ANOVA, $F_{3,42} = 0.36$, p = 0.778; Student's *t* test, $t_{14} = 4.88$, p = 0.0002). Data are presented as mean ± SEM, *n* values refer to the number of mice per group, *p < 0.05. (**B**) Object location memory test. (**i**) Scheme of the experiment. (**ii**) Satb2 cKO mice (*n* = 11) and control mice (*n* = 10) exhibited similar preference for the novel location over the familiar location at the 1 hr memory retention test (Student's *t* test, $t_{19} = 0.46$, p = 0.65). (**iii**) Satb2 cKO mice (*n* = 8) showed reduced preference for the novel location over the familiar location at the 24 hr memory retention test (Student's *t* test, $t_{14} = 2.89$, p = 0.011) compared to *Satb2*[flox/flox] mice (*n* = 8). The relative exploration time is expressed as a percent discrimination index (D.I. = ($t_{novel\ location}$ − $t_{familiar\ location}$) / ($t_{novel\ location}$ + $t_{familiar\ location}$) × 100%). Data are presented as mean ± SEM, *n* values refer to the number of mice, **p < 0.01. (**C**) Novel object recognition test. (**i**) Scheme of the experiment. (**ii**) Satb2 cKO mice (*n* = 10) and control mice (*n* = 9) exhibited a similar preference for the novel over the familiar object at the 1 hr memory retention test (Student's *t* test, $t_{19} = 1.11$, p = 0.28). (**iii**) Satb2 cKO mice (*n* = 8) spent

*Figure 2 continued on next page*

*Figure 2 continued*

less time exploring the novel object at the 24 hr memory retention test (Student's *t* test, $t_{14} = 3.0$, p = 0.009) compared to *Satb2*^flox/flox mice (*n* = 8). The relative exploration time is expressed as a percent discrimination index (D.I. = ($t_{novel\ object}$ − $t_{familiar\ object}$) / ($t_{novel\ object}$ + $t_{familiar\ object}$) × 100%). Data are presented as mean ± SEM, *n* values refer to the number of mice, *p < 0.05.

The following figure supplement is available for figure 2:

**Figure supplement 1.** Satb2 cKO mice show normal responses to electric foot shock.

---

blocked the induction of Satb2 by BDNF (*Figure 4E*). Furthermore, inhibition of mitogen/stress-activated kinase 1 (MSK1), a major regulator of activity- and experience-dependent synaptic adaptation downstream of MEK1/2 (*Corrêa et al., 2012*), had the same effect as the inhibition of MEK1/2 (*Figure 4F*), indicating that in hippocampal neurons BDNF up-regulates Satb2 through a pathway that requires ERK1/2 and MSK1.

Finally, we examined the temporal pattern of Satb2 induction by BDNF. Satb2 protein was increased at 6 hr, reached a maximum within 12–24 hr, and remained at this level for 48 hr following BDNF stimulation (*Figure 4—figure supplement 2A*). Loss of Satb2 occurred with a similar kinetic after antagonizing Trk receptor signaling in cultures previously stimulated with BDNF for 24 hr (*Figure 4—figure supplement 2B*). The relatively slow kinetics of Satb2 induction and elimination in hippocampal neurons is consistent with a potential role of Satb2 in slow, long-lasting adaptive neuronal processes (*Zagrebelsky and Korte, 2014*).

## Satb2 occupies active gene promoters

To gain insights into the molecular mechanisms by which Satb2 contributes to LTP maintenance and memory consolidation we mapped Satb2 genomic binding sites by ChIP-seq. Mouse primary hippocampal neurons were transduced with an AAV encoding V5-tagged Satb2 and a V5 antibody was used for chromatin immunoprecipitation. The specificity of the anti-V5 antibody was verified by ChIP-qPCR in non-transduced primary neurons (*Figure 5—figure supplement 1*). Out of 8414 Satb2 binding sites identified, 4496 were located within less than 1 kb distance from a transcriptional start site (TSS), indicative of Satb2 enrichment at proximal promoters (*Figure 5A and B*). Promoter sequences in eukaryotic genomes are marked by histone tail modifications associated with active or inactive state of the downstream gene (*Barski et al., 2007*; *Wang et al., 2008*). To examine the chromatin states of Satb2-bound promoters we compared our Satb2 ChIP-seq data with previously published datasets reporting genome-wide histone modifications and RNA Pol II recruitment in cortical neurons (GSE63271, GSE66701 and GSE21161) or in hippocampal tissue (GSE65159). Statistical testing of the overlap among the peaks of these datasets and our Satb2 ChIP-seq data revealed highly significant enrichment (adjusted p<0.00024) of active chromatin states (H4K16ac, H3K27ac, H3K4me1, H3K4me2, and H3K4me3 peaks) and PolII peaks at Satb2-bound promoters (*Figure 5C*, *Figure 5—figure supplement 2A*). Conversely, Satb2 genome occupancy and the Polycomb-associated H3K27me3 repressive mark showed no correlation (*Figure 5—figure supplement 2B*). GO enrichment analysis of the genes with Satb2 peaks within their promoters revealed significant over-representation of the following GO categories: 'transcription factor activity', 'transcription corepressor activity', 'chromatin remodelling complex', 'dendritic spine' and the KEGG pathway 'long-term potentiation' (*Figure 5—source data 1*)

Given that the majority of annotated gene promoters are associated with CpG islands (*Deaton and Bird, 2011*) we examined if Satb2 localizes to CpG islands. ChIP-seq tag distribution profiles revealed that Satb2 was deposited at CpGs associated with proximal promoters, intragenic (gene body associated) or intergenic CpGs (*Figure 5—figure supplement 3A–C*). Evidence suggests that most of the latter two CpG classes represent alternative promoters of nearby annotated genes or TSSs for non-coding RNAs (*Monteys et al., 2010*; *Wang et al., 2010*). Indeed, we found Satb2 enrichment on miRNA-associated CpGs (*Figure 5—figure supplement 3D*).

To further explore if Satb2 binds to miRNA TSSs, we used a recently developed miRNA promoter prediction method (*Marsico et al., 2013*) to assess potential enrichment of Satb2 on miRNA promoters. The miRNA TSS identification algorithm that we applied detected at least one TSS for about

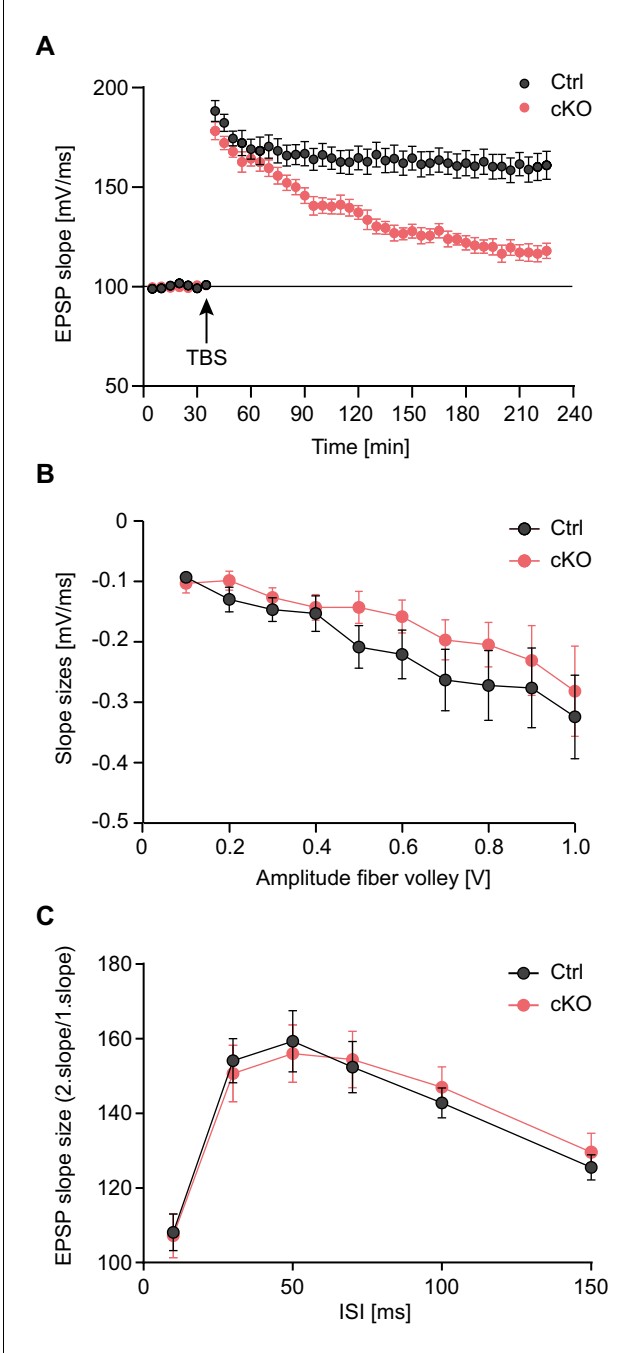

**Figure 3.** Late-LTP maintenance is impaired in Satb2 conditional mutants. (**A**) Schaffer collateral-CA1 late-LTP was significantly impaired in Satb2 cKO mice (Student's *t* test, $t_{28}$ = 4.92, p < 0.0001 for the interval 180–185 min post-theta burst stimulation, TBS). Shown are field EPSP slopes in *Satb2*flox/flox (Ctrl, *n* = 17 slices, 6 mice) vs. Satb2 cKO mice (cKO, *n* = 13 slices, 6 mice) recorded before and after TBS (100 Hz, repeated three times in a 10 s interval). Data are presented as mean ± SEM. (**B**) Input-output curves comparing the amplitudes of the presynaptic fiber volley to the field EPSP amplitude across a range of stimulation currents showed that basal synaptic transmission did not differ in hippocampal slices from Satb2 cKO mice (cKO, *n* = 17 slices, 6 mice) and littermate controls (Ctrl, *n* = 13 slices, 5 mice); Student's *t* test, p > 0.05 for all data points. Data are presented as mean ± SEM. (**C**) Paired-pulse facilitation studies across different inter-stimulus intervals revealed no difference between Satb2 cKO mice (cKO, *n* = 17 slices, 6 mice) and littermate controls (Ctrl, *n* = 13 slices, 5 mice); Student's *t* test, p > 0.05 for all data points. Data are presented as mean ± SEM.

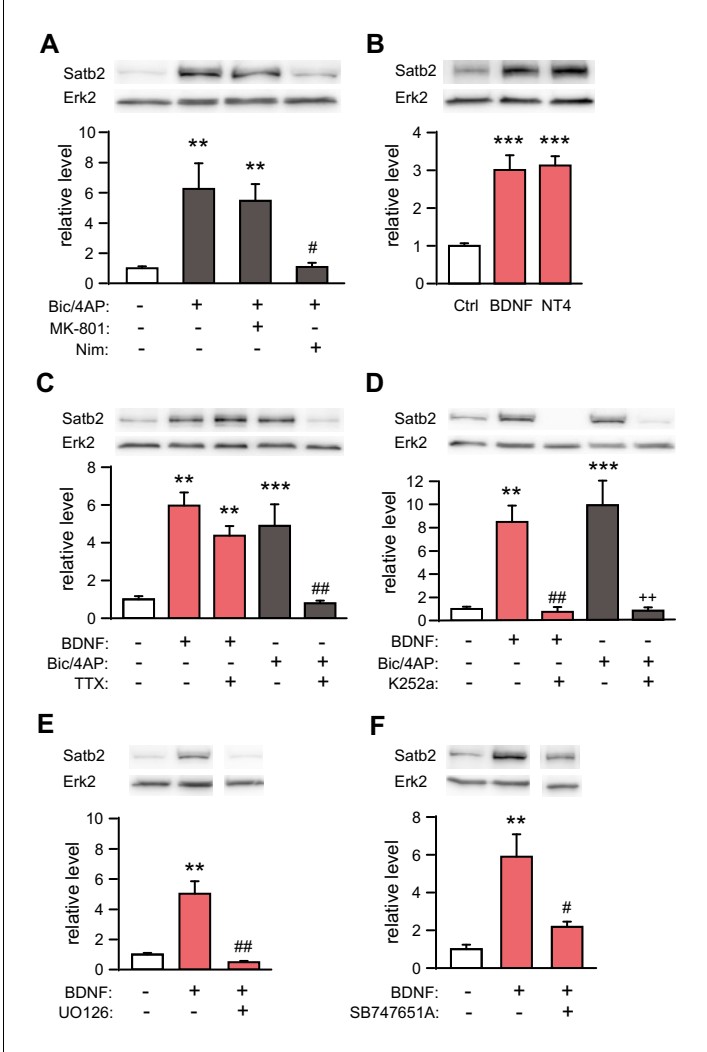

**Figure 4.** Synaptic activity and BDNF up-regulate Satb2 in primary hippocampal neurons. (**A**) Increased synaptic activity up-regulates Satb2 protein depending on calcium influx through L-type VGCC. Representative Western blot (top) and quantification (bottom) of the Satb2 protein level 24 hr after Bic/4AP treatment in the presence or absence of L-VGCC blocker nimodipine or NMDAR antagonist MK-801 ($n$ = 10, 6, 7, 5; ANOVA followed by Hochberg *post hoc* test; $F_{3,24}$ = 9.171, Ctrl vs. Bic/4AP, p = 0.002; Ctrl vs. Bic/4AP+MK-801, p = 0.006; Bic/4AP vs. Bic/4AP+Nim, p = 0.011). (**B**) BDNF and NT4 significantly increase Satb2 protein 24 hr after treatment. Representative Western blot image (top) and quantification of the Satb2 protein level (bottom) are shown, $n$ = 4; ANOVA followed by Tukey *post hoc* test, $F_{4,15}$ = 15.4, Ctrl vs. BDNF, p = 0.0004; Ctrl vs. NT4, p = 0.0002. (**C**) TTX application does not prevent Satb2 induction by BDNF but abolishes Satb2 up-regulation by synaptic activity. Representative image of immunoblot analysis (top) and quantification of the Satb2 protein level (bottom) are shown, $n$ = 3–7; ANOVA followed by Hochberg *post hoc* test, $F_{4,22}$ = 12.5, Ctrl vs. BDNF, p = 0.002, Ctrl vs. BDNF + TTX, p = 0.001, Ctrl vs. Bic/4AP, p = 0.0004, Bic/4AP vs. Bic/4AP+TTX, p = 0.002. Treatments with Bic/4AP also contained MK-801. (**D**) Treatment with the Trk inhibitor K252a completely blocks the up-regulation of Satb2 by both BDNF and Bic/4AP Representative image of Western blot (top) and quantification of the Satb2 protein level (bottom) are shown, $n$ = 4–6; ANOVA followed by Hochberg *post hoc* test, $F_{4,20}$ = 15.6 Ctrl vs. BDNF, p = 0.001, Ctrl vs. Bic/4AP, p = 0.0001, BDNF vs. BDNF+K252a, p = 0.002, Bic/4AP vs. Bic/4AP+K252a, p = 0.0002. Treatment with Bic/4AP also contained MK-801. (**E**) Blockade of the ERK1/2 signaling pathway with UO126 inhibits the induction of Satb2 by BDNF. Representative image of Western blot (top) and quantification of Satb2 protein levels (bottom) are shown, $n$ = 4; ANOVA followed by Tukey *post hoc* test, $F_{3,12}$ = 26.7, Ctrl vs. BDNF, p = 0.003; BDNF vs. BDNF+UO126, p = 0.001. (**F**) Inhibition of ERK1/2-downstream kinase MSK1 prevents BDNF-induced Satb2 up-regulation. Representative image of Western blot (top) and quantification (bottom) are shown, $n$ = 4–5; ANOVA followed by Hochberg *post hoc* test, $F_{2,11}$ = 11.4, Ctrl vs. BDNF, p = 0.002, BDNF vs. BDNF+SB747651A, p = 0.018. In (**A–F**), data are presented as mean ± SEM of the indicated number of experiments, $n$ values refer to

*Figure 4 continued on next page*

*Figure 4 continued*

the number of independent hippocampal cultures, *p < 0.05; **p < 0.01; ***p < 0.001, compared with Ctrl; #p <
0.05; ##p < 0.01; ###p < 0.001, compared with BDNF; +p < 0.05; ++p < 0.01; +++p < 0.001, compared with Bic/4AP.

The following figure supplements are available for figure 4:

**Figure supplement 1.** BDNF-induced Satb2 expression requires gene transcription.

**Figure supplement 2.** Time-course analysis of Satb2 expression after BDNF treatment.

---

82% of the miRBase-annotated miRNAs (*Marsico et al., 2013*). Furthermore, it is particularly suited
for detection of intronic miRNA promoters, which often act independently from the host gene pro-
moters (*Monteys et al., 2010*). Our analysis revealed significant association of Satb2 with miRNA
promoters (*Figure 5D*).

To confirm that endogenous Satb2 in the adult hippocampus binds to the same regions identified
by ChIP-seq in hippocampal cultures we performed ChIP-qPCR using a Satb2-specific antibody and
chromatin derived from control and Satb2 cKO CA1 hippocampal tissue. The results revealed Satb2
enrichment at various identified target promoters and/or Satb2 binding sites in chromatin samples
from control but not Satb2 cKO mice, thus validating the *in vitro* Satb2 genomic binding patterns
(*Figure 5—figure supplement 4*).

Taken together, our results provide evidence for association of Satb2 with active promoter regu-
latory sequences in the genome of hippocampal neurons, including miRNA promoters, suggesting a
potential role of Satb2 in the transcription of active neuronal chromatin.

## Satb2 determines the expression of protein-coding genes and miRNAs linked to learning and memory in the CA1 hippocampal field

To study the role of Satb2 in transcriptional control *in vivo*, we performed global transcriptome pro-
filing analyses (RNA-seq and small RNA-seq) using CA1 hippocampal tissue from Satb2 cKO mice
and littermate controls. RNA-seq analysis identified a number of protein-coding genes that were dif-
ferentially expressed between control and Satb2 cKO mice (25 up-regulated and 15 down-regulated,
*Figure 6—source data 1*). Amongst them we found genes that have previously been identified as
highly relevant for learning and memory or directly implicated in memory formation such as *Adra2a*,
*Penk*, *Htr5b*, and *Ghsr* (*Diano et al., 2006*; *Galeotti et al., 2004*; *Ghersi et al., 2015*; *Peppin and
Raffa, 2015*). Pathway analysis revealed significant enrichment (p=0.018) of the 'neuroactive ligand-
receptor interaction pathway' amongst the regulated genes. The differential expression of selected
genes, including *Adra2a*, *Penk*, *Htr5b*, and *Ghsr* was validated by real-time qPCR (*Figure 6—figure
supplement 1*). Notably, Satb2 was bound to the promoters of the majority of the identified differ-
entially expressed genes as shown by Satb2 ChIP-seq (*Figure 6—source data 1*). We also investi-
gated differential splicing between the CA1 transcriptomes of Satb2 cKO mice and littermate
controls, since global de-regulation of RNA-splicing has been linked to impaired synaptic plasticity
and memory function (*Benito et al., 2015*). However, no major difference in RNA-spicing was
observed comparing Satb2 cKO and control mice (*Figure 6—source data 2*).

Next, we performed small RNA-seq analysis to examine miRNA transcriptome changes in the
CA1 hippocampal area of Satb2 cKO vs control mice. We detected the expression of 476 miRNAs in
the CA1 tissue, similar to the number reported for this hippocampal subfield in a previous study
(*Stilling et al., 2014*). Of these 464 miRNAs 43.9% showed significant differential expression
between Satb2 cKO and control mice (*Figure 6A*, *Figure 6—source data 3*). Principal component
analysis, a method for visualizing gene expression patterns, revealed a clear separation between
Satb2 cKO and control samples (*Figure 6B*). Furthermore, hierarchical clustering using a Pearson
correlation-based method demonstrated two main clusters of miRNAs (up- and down regulated)
based on their expression levels in the CA1 field of Satb2 cKO vs control mice (*Figure 6C*). The dif-
ferential expression of selected miRNAs in the CA1 tissue of control vs. Satb2 cKO mice was vali-
dated by qPCR (*Figure 6—figure supplement 2*). 44.4% of the miRNAs found to be deregulated in
Satb2 cKO mice had at least one predicted promoter occupied by Satb2 as demonstrated by Satb2
Chip-seq data (*Figure 6—figure supplement 3* and *Figure 6—source data 4*). Moreover, *in vivo*

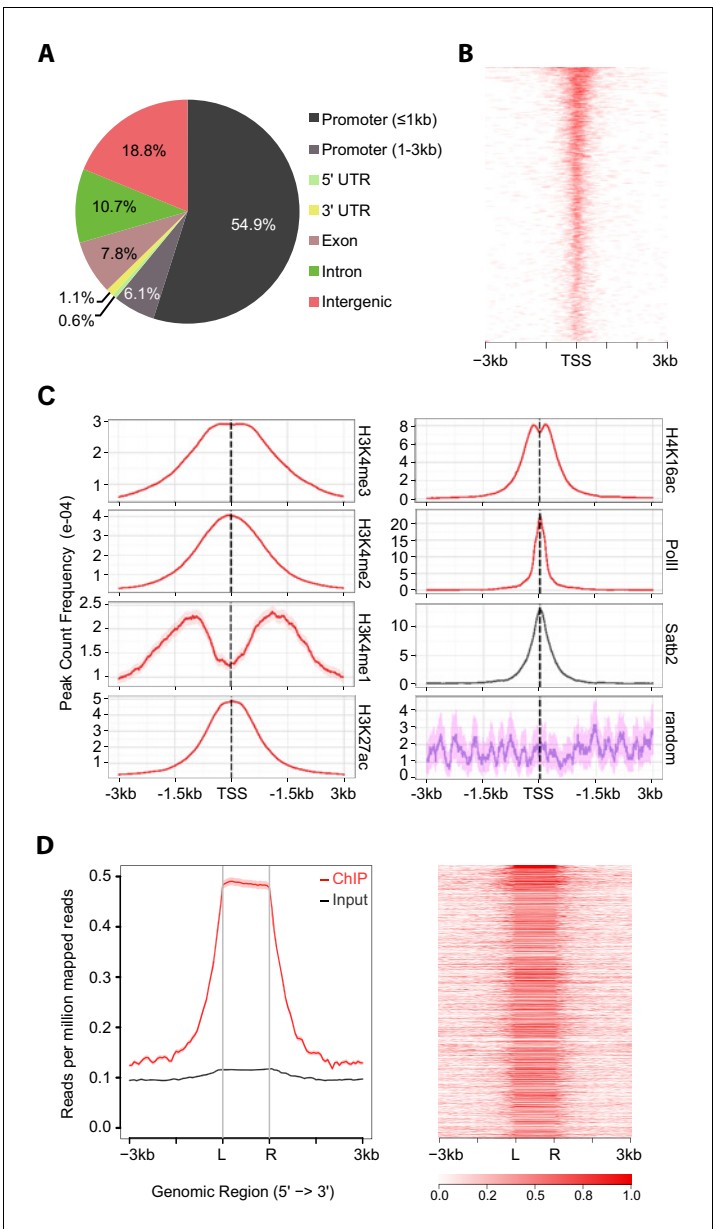

**Figure 5.** Satb2 binding sites are enriched on active gene promoters including miRNA promoters. (**A**) Pie-chart illustrating the genomic annotation of Satb2 binding sites. (**B**) Heatmap of Satb2 binding to TSS (±3 Kb) regions. (**C**) Average profiles of H4K16ac, H3K27ac, H3K4me1, H3K4me2, H3K4me3 and PolII peaks (GEO: GSE63271, GSE66701, GSE21161, and GSE65159) at Satb2 bound promoters. (**D**) Average tag density profiles (ChIP/Input, left panel) and heat map depicting Satb2 ChIP-seq tag density at predicted miRNA promoter regions (right panel). 'L' – 5' left, 'R' – 3' right of the miRNA promoters. The tick marks represent distance of −3 kb, −1.5 kb, +1.5 kb, +3 kb relative to the miR promoters.

The following source data and figure supplements are available for figure 5:

**Source data 1.** Major GO terms and KEGG pathways, revealed by ChIP-Enrich bioinformatics tool, found to be enriched among the genes having Satb2 peaks within their promoters.

**Figure supplement 1.** ChIP-qPCR validation of Satb2 enrichment at various identified target regions using chromatin from AAV-Satb2-V5-transduced and non-transduced primary hippocampal neurons.

**Figure supplement 2.** Satb2 binding sites are enriched on active gene promoters and do not correlate with the Polycomb-associated H3K27me3 repressive mark (PcR).

*Figure 5 continued on next page*

*Figure 5 continued*

**Figure supplement 3.** Satb2 is deposited at CpGs.

**Figure supplement 4.** ChIP-qPCR validation of Satb2 targets *in vivo*.

Chip-qPCR analysis using an antibody against Satb2 and chromatin from CA1 tissue demonstrated that Satb2 can bind to the miR-22 promoter (*Figure 5—figure supplement 4*). Importantly, amongst the miRNAs identified as deregulated in Satb2 cKO mice there were miRNAs with well-documented synaptic regulatory functions, including miR-125b, miR-132, miR-212, miR-124 (*McNeill and Van Vactor, 2012*), or miRNAs which have been directly implicated in learning and memory, such as miR-132 or miR-124 (*Hansen et al., 2010*; *Malmevik et al., 2016*).

Collectively, the transcriptome profiling results provide evidence for an important role of Satb2 as a regulator of hippocampal miRNA expression.

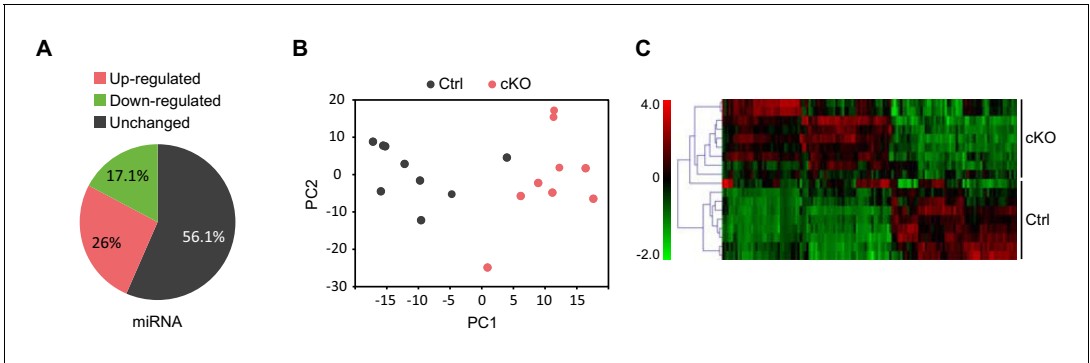

**Figure 6.** Satb2 regulates miRNA expression in CA1 hippocampal area. (**A**) Pie-chart showing the percentage of differentially expressed miRNAs (up- and down-regulated, *Figure 6—source data 1*) in the CA1 region of Satb2 cKO mice vs. littermate controls as assessed by small RNA-seq analysis (base mean above 10 counts, 1.5-fold change, and adjusted p < 0.05). (**B**) PCA plot of miRNA counts analyzed by small RNA-seq of Satb2 cKO vs. control CA1 hippocampal tissue. The first two PCs explained 33.3% and 21.4% of the variance, respectively. (**C**) Heat map from hierarchical clustering of differentially expressed miRNAs (base mean above 100 counts) in the CA1 hippocampal area of Satb2 cKO mice (*n* = 9) vs. littermate controls (*n* = 9).

The following source data and figure supplements are available for figure 6:

**Source data 1.** Differentially expressed genes between Satb2 cKO mice and *Satb2*[flox/flox] littermate controls in the CA1 region as assessed by RNA-seq.

**Source data 2.** No major significant differences in splicing were observed between the CA1 region of Satb2 cKO mice (*n* = 9) and littermate controls (*n* = 9).

**Source data 3.** List of differentially expressed miRNAs between Satb2 cKO mice and littermate controls (*Satb2*[flox/flox] mice) in the CA1 region as assessed by small RNA-seq analysis (base mean above 10 counts, 1.5-fold change, and adjusted p < 0.05).

**Source data 4.** List of differentially expressed miRNAs in the CA1 region of Satb2 cKO mice vs. littermate controls that contain Satb2 peak(s) within their promoter.

**Figure supplement 1.** Validation of the differential expression of selected genes by qPCR.

**Figure supplement 2.** Validation of the differential expression of selected miRNAs by qPCR.

**Figure supplement 3.** Satb2 binds to miRNA's promoters.

# Re-expression of Satb2 into adult hippocampus rescues both a decrease in Arc protein and fear memory deficits

Among the miRNAs up-regulated in the CA1 of Satb2 cKO mice, 24 miRNAs were predicted to target the 3'UTR of 'activity–regulated cytoskeletal associated protein' (Arc) by at least one of the miRNA-target prediction tools TargetScan, miRanda, and PITA (*Figure 7—source data 1*). Moreover, regulation of Arc translation by multiple miRNAs has already been demonstrated in primary hippocampal neurons (*Wibrand et al., 2012*). Given the crucial role of Arc in experience-dependent synaptic plasticity and long-term memory (*Korb and Finkbeiner, 2011*; *Shepherd and Bear, 2011*) and the deficits in LTP and fear memory observed in Satb2 cKO mice, we examined whether Arc protein is altered *in vivo* in the CA1 of Satb2 mutants. Indeed, immunoblotting analysis revealed significantly reduced Arc protein level in Satb2 cKO mice compared to littermate controls (*Figure 7A*). At the same time, Arc mRNA level was not altered (*Figure 7B*).

To investigate if restoring Satb2 expression can rescue the observed reduction in Arc level and/or the impaired long-term fear memory of Satb2 cKO mice, we used rAAV-mediated gene delivery to re-introduce Satb2 into the hippocampus of *Satb2* mutants. Viral vectors encoding either Satb2 (rAAV8-hSyn-*Satb2*-V5) or GFP (rAAV8-hSyn-*GFP*) were injected into the dorsal hippocampus of Satb2 cKO mice four weeks prior to immunoblotting analysis or contextual fear conditioning training. Littermate *Satb2*flox/flox mice, injected with rAAV8-hSyn-*GFP*, served as controls. Immunohistochemistry analysis confirmed robust and specific expression of V5-tagged Satb2 in the hippocampal CA1 region of conditional mutants after stereotaxic injection of rAAV8-hSyn-*Satb2*-V5 (*Figure 7C*). Immunoblotting results revealed that rAAV8-hSyn-*Satb2*-V5 injection was able to restore Arc protein levels in the CA1 area of Satb2 cKO mice to control levels (*Figure 7D*). Moreover, the reinstatement of Satb2 in the dorsal hippocampus not only reversed the decreased Arc protein level, but also rescued the long-term contextual fear memory impairment. We observed indistinguishable freezing behavior between control and rAAV8-hSyn-*Satb2*-V5-transduced Satb2 cKO mice 24 hr after training (*Figure 7E*). By contrast, Satb2 cKO mice injected with rAAV8-hSyn-*GFP* showed significantly reduced freezing, reproducing our initial findings with un-injected animals (*Figure 2A*).

## Discussion

Our results establish Satb2 as an important determinant of memory consolidation in the adult hippocampus. At molecular level we show in primary hippocampal neurons that calcium influx through L-type VGCC as well as BDNF up-regulate Satb2 in the nucleus, where it binds to promoters of coding and non-coding loci. Altered expression of Satb2-dependent miRNAs on a genome-wide scale is likely to cause changes in the posttranscriptional regulation of synaptic plasticity proteins, exemplified by Arc. Consistent with these molecular mechanisms, the lack of Satb2 in the forebrain causes impairments in late-phase LTP and long-term memory in adult mice.

Given the potential of Satb2 to mediate DNA looping and local association of gene promoter regions (*Zhou et al., 2012*), our findings suggest that some of the key functions of the hippocampus may depend on changes in the higher-order chromatin architecture. Long-range looping interactions between promoters and distal regulatory elements are considered to recruit transcription factors and chromatin-modifying complexes to gene promoters (*Bharadwaj et al., 2014*; *Sanyal et al., 2012*). Our data demonstrating specific binding of Satb2 to active coding or non-coding gene promoters, co-occupied by RNA Pol II, are consistent with this model. Based on our results in primary hippocampal cultures, it can be proposed that this type of higher-order chromatin rearrangement is an activity- and BDNF-dependent process that involves changes in Satb2 expression levels. Of note, the reported schizophrenia risk allele for SATB2 (rs6704641) is intronic and likely affects SATB2 mRNA levels rather than protein function (Schizophrenia Working Group of the *Schizophrenia Working Group of the Psychiatric Genomics Consortium, 2014*). This finding strengthens the hypothesis that quantitative changes in Satb2, which otherwise is expressed in all CA1 pyramidal neurons, impact on Satb2-chromatin interactions. The relevance of chromatin looping in cognition or psychiatric diseases has already been shown for the regulation of individual genes: the NMDA receptor locus GRIN2B46 (*Bharadwaj et al., 2014*), and two schizophrenia-risk genes, encoding the GABA synthesis enzyme GAD1 and the calcium channel alpha subunit CACNA1C (*Nestler et al., 2016*; *Rajarajan et al., 2016*). Noteworthy, the homologue of Satb2, Satb1, a known chromatin structure organizer, affects the formation of dendritic spines and modulates the

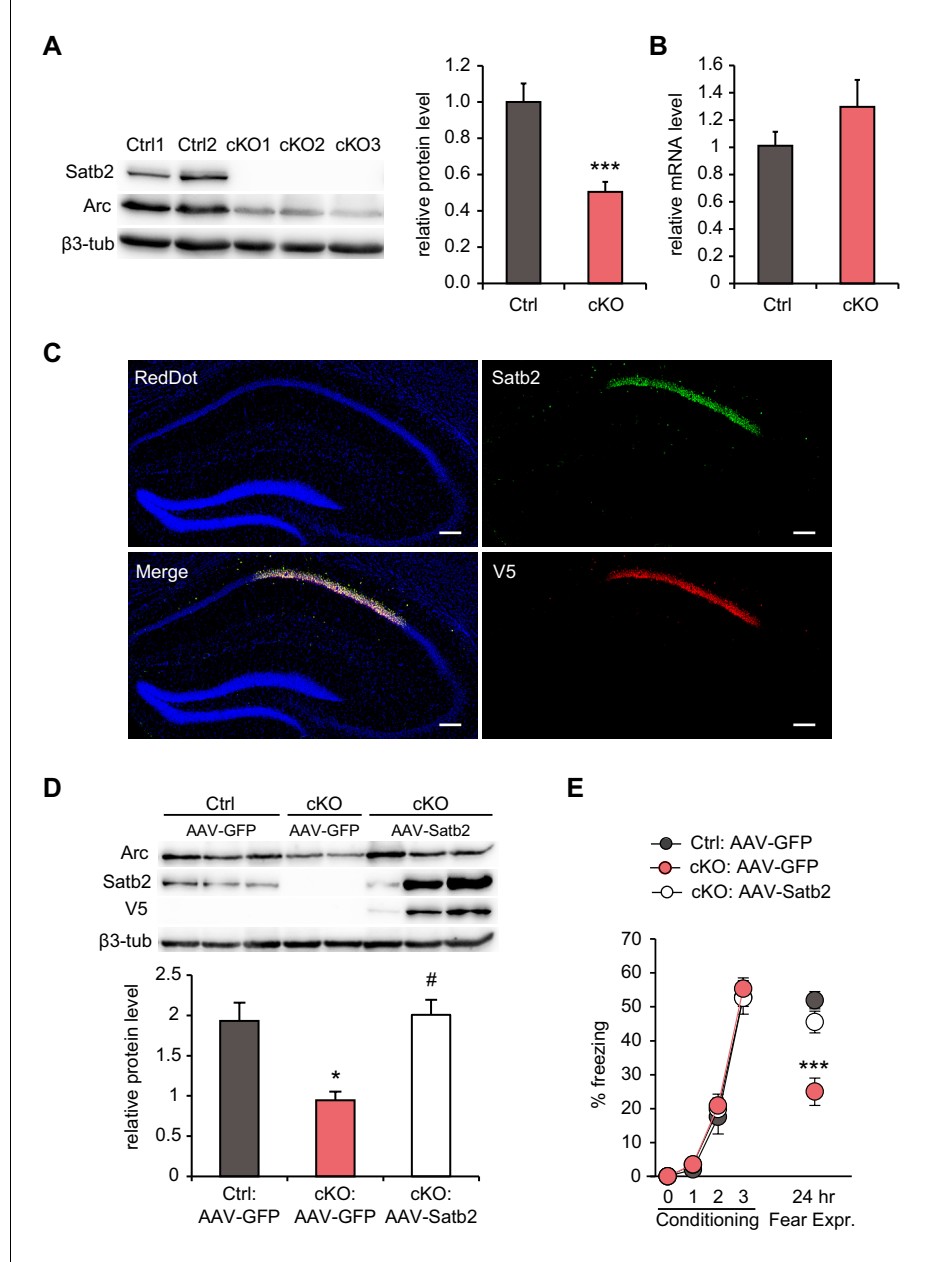

**Figure 7.** Hippocampal Satb2 re-expression rescues Arc levels and long-term fear memory deficits. (**A**) Representative Western blot (left) and quantification (right) of Arc protein level in the CA1 field of Satb2 cKO mice and littermate controls. β3-tubulin was used as a loading control (Ctrl, $n = 9$; cKO, $n = 11$; Student's $t$ test, $t_{18} = 4.52$, p = 0.0003). Data are presented as mean ± SEM, $n$ values refer to the number of mice per group, ***p < 0.001 compared to Ctrl. (**B**) qPCR quantification of Arc mRNA level in the CA1 field of Satb2 cKO mice and littermate controls (Ctrl, $n = 3$; cKO, $n = 4$; Student's $t$ test, $t_5 = 1.14$, p = 0.303). Data are presented as mean ± SEM, $n$ values refer to the number of mice used. (**C**) Satb2/V5-immunoreactivity in the CA1 hippocampal area of Satb2 cKO mice after stereotaxic injection of rAAV-hSyn-*Satb2-V5* into the dorsal hippocampus. Nuclei were counterstained with RedDot. Representative images are shown. Scale bar: 150 µm. (**D**) Representative Western blot (left) and quantification (right) of Arc protein level in the CA1 field of *Satb2*<sup>flox/flox</sup> mice injected with rAAV-hSyn-*eGFP* (Ctrl:AAV-*GFP*, $n = 7$), Satb2 cKO mice injected with rAAV-hSyn-*eGFP* (cKO:AAV-*GFP*, $n = 3$) and Satb2 cKO mice injected with rAAV-hSyn-*Satb2-V5* (cKO:AAV-*Satb2*, $n = 6$). β3-tubulin was used as a loading control. Re-expression of Satb2 in the dorsal hippocampus rescued the reduction in Arc protein in the CA1 area of Satb2 cKO mice, bringing it up to control levels (ANOVA followed by Fischer LSD *post hoc* test, $F_{2,13} = 5.011$, cKO:AAV-*GFP* vs. Ctrl:AAV-*GFP*, p = 0.014, cKO:AAV-*Satb2* vs. cKO:AAV-*GFP*, p = 0.011, Ctrl:AAV-*GFP* vs. cKO:AAV-*Satb2*, p = 0.793). Data are presented as mean ± SEM, $n$ values refer to the number of mice used, *p < 0.05 compared

*Figure 7 continued on next page*

*Figure 7 continued*

with Ctrl:AAV-*GFP*, #p < 0.05, compared with cKO:AAV-*GFP*. (E) In a contextual fear conditioning test, Ctrl:AAV-*GFP* (n = 9), cKO:AAV-*GFP* (n = 9) and cKO:AAV-*Satb2* (n = 14) mice showed similar levels of freezing during the fear-acquisition phase (repeated measures ANOVA, $F_{6,87}$ = 0.12, p = 0.99). Freezing behavior, analyzed 24 hr after the training, was significantly impaired in cKO:AAV-*GFP* mice, however the fear memory deficit was completely rescued in cKO:AAV-*Satb2* mice (ANOVA followed by Fischer LSD *post hoc* test, $F_{2,29}$ = 12.8, cKO:AAV-*GFP* vs. Ctrl:AAV-*GFP*, p = 0.00005, cKO:AAV-*Satb2* vs. cKO:AAV-*GFP*, p = 0.0004, Ctrl:AAV-*GFP* vs. cKO:AAV-*Satb2*, p = 0.22). Data are presented as mean ± SEM, *n* values refer to the number of mice per group, ***p < 0.001 compared with cKO:AAV-*Satb2* and Ctrl:AAV-*GFP*.
The following source data is available for figure 7:

**Source data 1.** List of miRNAs predicted to target mouse Arc 3'UTR by the bioinformatics tools TargetScan, PITA and miRanda.

expression of multiple immediate early genes in the cortex (*Balamotis et al., 2012*), thus corroborating the importance of Satb family members in neuronal plasticity.

We found that a large fraction of all miRNAs expressed in the adult CA1 hippocampal field is deregulated when Satb2 is genetically ablated. This establishes Satb2 as a novel regulator of the miRNA transcriptome in CA1 pyramidal neurons. Notably, several miRNAs with altered expression in Satb2 mutants, e.g. miR-124, miR-125b, miR-132, miR-212, miR-381, miR-326, miR-19b have already been shown to affect the translation of proteins important in various aspects of synaptic plasticity or memory formation (*Aksoy-Aksel et al., 2014*; *Ryan et al., 2015*). At synaptic sites translation of locally synthesized proteins is at least in part repressed by miRNAs. The miRNA machinery interacts with fragile X mental retardation protein (FMRP), which acts as translational repressor. Intriguingly, almost all of the miRs found to be associated with FMRP (miR-100, miR-124, miR-125a, miR-127, miR-128, miR-132, miR-143) (*Edbauer et al., 2010*) were up-regulated in Satb2 mutants. This finding implies a decreased expression of synaptic proteins after Satb2 ablation and it also provides interesting candidates for future investigations. In our study, we have already demonstrated a decreased level of Arc in the CA1 of Satb2 cKO mice. The identification and validation of additional Satb2-dependent miRNA–mRNA interactions *in vivo* in the adult brain will be important to elucidate the mechanisms, by which Satb2 regulates memory formation. Recent studies have demonstrated enhanced spatial learning and working memory capacity after inhibition of miR-124 in the hippocampus or restored spatial memory, social interaction and LTP impairments in adult mice carrying a null mutation for EPAC protein (*Malmevik et al., 2016*; *Yang et al., 2012*). Our data showing up-regulation of miR-124 and impaired LTP and long-term memory in Satb2 mutants are in full agreement with these reports. Interestingly, amongst the miRNAs deregulated in Satb2-deficient CA1 we also found miRNAs shown to be deregulated in schizophrenia animal models or in schizophrenia patients (*Beveridge and Cairns, 2012*).

Although the precise mechanism(s) of Satb2-dependent miRNA regulation remain to be determined, two lines of argument support the view that Satb2 has a direct influence on miRNA transcription. First, we find individual miRNAs to be up- as well as down-regulated, arguing against a general effect of Satb2 on miRNA biogenesis or processing by the regulation of Dicer as has been reported for BDNF (*Huang et al., 2012*). Second, we find that miRNA promoter elements are bound by Satb2 in hippocampal neurons. Our ChIP-seq data in primary cultures demonstrate a potential of Satb2 to occupy both sets of promoters (protein-coding and non-coding). The preferential regulation of miRNA expression over protein-coding genes that we observed *in vivo* might for example be explained by the presence of Satb2 co-interactors in mature neurons that determine selective binding and control of miR promoters.

In conclusion, we provide evidence that Satb2 is required for synaptic plasticity and long-term memory formation in the adult CNS, likely via the regulation of miRNAs and protein-coding genes controlling synaptic structure and function. Our findings offer a plausible mechanism explaining the intellectual disability and severe learning difficulties observed in SAS patients. Furthermore, the Satb2 function in the adult brain unsuspectedly interconnects several individual components that have been discussed in the context of various psychiatric syndromes: BDNF signaling, epigenetic

chromatin modifications, miRNA dysregulation and cognitive dysfunction. Therefore, in the future, it will be intriguing to establish $Satb2^{flox/flox}$::Camk2a-Cre conditional mutant as an animal model of neuropsychiatric diseases.

## Materials and methods

### Animals

Mice carrying an allele of *Satb2* in which exon 4 is flanked by loxP sites ($Satb2^{flox/flox}$) were generated by microinjection of ES cells carrying a $Satb2^{tm1a(KOMP)Wtsi}$ Knockout First (Promoter driven) allele (clone *Satb2*_G07, JM8.N4 subline; KOMP repository) into blastocysts from albino C57BL/6J donor mice. The germline-positive mice were further crossed with FLPo deleter mice (RRID:MMRRC-032247-UCD) (*Kranz et al., 2010*) to excise the FRT cassette and to establish a conditional allele. To generate *Satb2* conditional mutants $Satb2^{flox/flox}$ mice were crossed with *Camk2a-Cre* mice (*Minichiello et al., 1999*) on a C57BL/6 background. In all experiments, mice that carry the floxed exon 4 but do not express the *Cre* transgene ($Satb2^{flox/flox}$) were used as littermate controls. Unless otherwise stated, adult male mice at the age of 3–4 months were used for behavioral and molecular analyses. All experimental procedures were approved by the Austrian Animal Experimentation Ethics Board (Bundesministerium für Wissenschaft und Verkehr, Kommission für Tierversuchsangelegenheiten).

### Antibodies

*Primary antibodies.* Satb2 ab92446 (AB_10563678), V5 tag antibody ChIP grade ab15828 (AB_443253), Satb2 ab34735 (AB_2301417), Ctip2 ab18465 (AB_2064130), Tbr1 ab31940 (AB_2200219) were purchased from Abcam (Cambridge, MA); Erk2 sc-154-G (AB_631459), Arc sc-17839 (AB_626696), Cux1 sc-13024 (AB_2261231) were obtained from Santa Cruz (Dallas, TX); Arc 156 003 (AB_887694) was ordered from Synaptic Systems (Germany); V5 epitope tag antibody R960-25 (AB_2556564) was obtained from Thermo Fisher Scientific (Waltham, MA); Satb2 AMAb90682 was purchased from Atlas Antibodies (Sweden); Wfs1 11558–1-AP (AB_2216046) was ordered from Proteintech (Rosemont, IL); and beta-III Tubulin antibody NB100-1612 (AB_10000548) was obtained from Novus Biologicals (UK).

*Secondary antibodies.* Goat anti-mouse Alexa-488 A11001 (AB_2534069), donkey anti-mouse Alexa-488 A21202 (AB_2535788), goat anti-rabbit Alexa-555 A21428 (AB_10561552), donkey anti-rabbit Alexa-488 A21206, donkey anti-rat Alexa 488 A21208 (AB_2535792) were all purchased from Thermo Fisher Scientific; and goat anti-mouse CF633 20121 (AB_10854245) was obtained from Biotium (Fremont, CA).

### Primary hippocampal culture

Hippocampi were dissected from newborn C57BL/6J mice at postnatal day P0 to P1. Hippocampal tissue was trypsinized and dissociated by trituration as described previously (*Kaech and Banker, 2006*). Neurons were plated on 35 mm tissue culture dishes or on 18 mm coverslips, previously coated with poly-L-ornithine (Sigma, St. Louis, MO) and laminin (Thermo Fischer Scientific), at a density of $8\times10^4$ cells/cm$^2$. Cells were cultured in Neurobasal medium (Thermo Fisher Scientific) containing B-27 supplement (Thermo Fisher Scientific), 100 mg/ml penicillin G and 60 mg/ml streptomycin sulfate in a humidified atmosphere of 5% $CO_2$ at 37°C for 8–10 days. Medium was replaced 1.5 hr after plating; thereafter one third of the culture medium was replaced with fresh medium at day in vitro (DIV) 3 or 4. Glial cell proliferation was inhibited by adding 5 µM cytosine arabinoside (Sigma) to the culture medium at DIV1. Cells were collected for immunoblotting analysis 24 hr after treatment. The following growth factors were used: BDNF (50 ng/ml, Peprotech, Rocky Hill, NJ), NT4/5 (50 ng/ml, a kind gift from Amgen-Regeneron). The following pharmacological substances were applied as described in the Results section: 50 µM bicuculline (Tocris Bioscience, UK), 500 µM 4-aminopyridine (Tocris Bioscience). Pharmacological inhibitors were added 1 hr prior to treatments with BDNF or Bic/4AP in the following concentrations: 10 µM nimodipine (Tocris Bioscience), 10 µM MK-801 (Tocris Bioscience), 10 µM SB202190 (Cell Signaling, Danvers, MA), 10 µM UO126 (Cell Signaling), 0.2 µM K252a (Santa Cruz), 333 ng/ml actinomycin D (Sigma) and 20 µM SB747651A (Axon Medchem, The Netherlands). Control cultures were treated with equal volumes of vehicle.

## Immunoblotting

Extracts of total protein were prepared by lysis of neurons on the cell culture dish in 2 x Roti-Load sample buffer (Carl Roth, Germany). Western blotting was performed as described previously (*Loy et al., 2011*). Membranes were blocked with 5% milk powder in TBST (0.1% Tween 20 in TBS) for 1 hr and then incubated overnight at 4°C with the corresponding primary antibodies, diluted in blocking solution. After incubation with HRP-coupled secondary antibodies, the blots were developed using ECL reagent (GE Healthcare, Chicago, IL). Blots were either developed using CP100 photo-developer (Agfa, Belgium) or FUSION-FX7 chemiluminescence detection system (Vilber Lourmat, France). Quantification of protein expression was carried out either by using ImageJ (RRID: SCR_003070) or FUSION-CAPT image analysis software in the linear range of detection.

## Immunohistochemistry

Mice were anesthetized and transcardially perfused with 4% PFA (w/v) in PBS (pH 7.4). Brains were removed, post-fixed for 2 hr in 4% PFA, cryo-protected in 30% (w/v) sucrose at 4°C and embedded in Tissue-Tek medium (Sakura Finetek Europe, The Netherlands). Free-floating cryosections (40 µm) were washed three times in Tris-Buffered Saline (TBS), permeabilized in 0.3% (v/v) Triton-X-100 in TBS for 5 min and incubated with blocking solution (10% (v/v) normal serum, 1% (w/v) BSA, 0.3% Triton-X-100 in TBS) for 2 hr. Sections were incubated with primary antibodies overnight at 4°C. After three washes in 0.025% Triton-X-100 in TBS sections were incubated with corresponding secondary antibodies. Sections were counterstained with Hoechst 33258 (H-3569, Molecular Probes, Grand Island, NY) for 5 min or with RedDot2 (40061, Biotium, Fremont, CA) for 20 min. After three washes in TBS sections were mounted with Roti-Mount FluorCare mounting medium (Carl Roth, Germany). Images were acquired using a LSM 510/Axiovert200M microscope (Carl Zeiss, Germany).

For cresyl violet staining free-floating brain sections (40 µm) were mounted on glass slides and incubated in cresyl violet (0.1 M sodium acetate, 2% acetic acid, 0.02 M cresyl violet acetate, Sigma, St. Louis, MO, in distilled water, pH 3.5).

## AAVs, viral transductions and stereotaxic injections

AAV8-hSyn-*eGFP* and AAV8-hSyn-*GFP-Cre* were purchased from UNC Gene Therapy Vector Core. *Satb2* coding sequence was *V5*-tagged and cloned into pAAV-hSyn-WPRE vector. AAV-hSyn-*Satb2-V5*, serotype 8, was generated by SignaGen Laboratories, Rockville, MD.

Hippocampal neurons were transduced with AAVs (AAV8-hSyn-*GFP-Cre* or AAV8-hSyn-*Satb2-V5*) at a multiplicity of infection $1.5 \times 10^5$ at DIV4 and used for ChIP-seq analysis at DIV11.

AAV8-hSyn-*eGFP* and AAV8-hSyn-*Satb2-V5* were delivered by stereotaxic injection into the dorsal hippocampus of 12 week old male mice (Satb2 cKO and control littermates). A burr hole was drilled into the skull, and 1 µl of viral stock ($10^{12}$–$10^{13}$ vg/ml) was injected bilaterally at a rate of 100 nl/min. The following coordinates (relative to Bregma) were used: antero-posterior, −2.1 mm; medio-lateral, ± 1.5 mm; dorso-ventral, −1.5 mm from the skull surface. The needle was left in place for 5 min after the injection. The mice were allowed to recover for four weeks after stereotaxic injections. Infection efficiencies of AAVs were determined by immunohistochemistry using antibodies to Satb2 or V5-tag or by analyzing the fluorescence of eGFP.

## RNA isolation and RT-qPCR

Total RNA was isolated from primary hippocampal cultures or dissected CA1 hippocampal tissue using TRIzol reagent (Thermo Fisher Scientific). cDNA was synthesized following the High-Capacity cDNA Reverse Transcription Kit protocol (Thermo Fisher Scientific). qPCR was performed using Fast SYBRGreen Master Mix (Thermo Fisher Scientific). All reactions were run in duplicates. The relative expression values were determined by normalization to *Gapdh* transcript levels and calculated using the ΔΔCT method (*Pfaffl, 2001*). Primers used for RT-qPCR analysis are listed in *Supplementary file 1*. Real-time PCR analysis of miRNAs was carried out using the miScript PCR System (Qiagen, Germany) following the manufacturer's instructions.

## RNA-seq

RNA was isolated from dissected CA1 hippocampal tissue using TRIzol (Thermo Fisher Scientific). Library preparation and cluster generation for mRNA and small RNA sequencing was performed

according to Illumina standard protocols (TruSeq, Illumina, San Diego, CA). Libraries were quality-controlled and quantified using a Nanodrop 2000 (Thermo Fisher Scientific), Agilent 2100 Bioanalyzer (Agilent Technologies, Santa Clara, CA) and Qubit (Thermo Fisher Scientific).

Base calling from raw images and file conversion to fastq-files was achieved by Illumina pipeline scripts. Subsequent steps included quality control (FastQC, www.bioinformatics.babraham.ac.uk/projects/fastqc/), mapping to reference genome (mm10, STAR aligner v2.3.0 (*Djebali et al., 2012*), non-default parameters), read counting on genes or exons (HTSeq, http://www-huber.embl.de/users/anders/HTSeq, mode: intersection-non-empty) and differential gene expression (DESeq2_1.4.5 [*Love et al., 2014*]) bio-statistical analysis.

PCA and distance heatmaps were generated in R following instructions in the vignette for DESeq2. A threshold cutoff of 1.5-fold change and adjusted (Benjamini-Hochberg) p value < 0.05 was applied.

Functional annotation and pathway analysis were carried out using the Database for Annotation, Visualization and Integrated Discovery (DAVID, RRID:SCR_003033) (*Huang et al., 2009*).

## ChIP-seq

AAV8-hSyn-*Satb2-V5*-transduced DIV11 primary hippocampal neurons were cross-linked with formaldehyde at 1% final concentration for 10 min at room temperature, and chromatin was prepared using the Zymo-Spin ChIP kit (Zymo Research, Irvine, CA) following manufacturer's instructions. Sonication was performed at high power setting for 40 cycles (30 s ON, 30 s OFF) using a Bioruptor Plus (Diagenode Inc., Denville, NJ), yielding fragment size range of 200–700 bp. ChIP assays were performed in triplicates using 20 µg of chromatin and 10 µg of anti-V5 tag antibody (ab15828). IgG (12–370, Millipore, Billerica, MA) was included as a negative control. ChIP DNA was purified using ChIP DNA Clean and Concentrator (Zymo Research) and the relative abundance of a control region in V5-immunoprecipitated DNA was quantified by qPCR with sequence-specific primers. DNA libraries (Satb2-ChIP and Input control DNA) were prepared and sequenced on a HiSeq 2000 sequencer (Illumina). Sequencing resulted in 39,309,191 and 88,731,003 high quality filtered 20–50 bp single end reads for ChIP- and Input DNA respectively. These reads were aligned to the mm10 mouse genome using the BWA short read aligner (*Li and Durbin, 2009*) (version 0.7.12). The aligned reads were filtered to keep only uniquely mapping reads (ChIP: 15,507,953 Input: 51,700,265). A cross-correlation analysis was performed to assess quality metrics (Phantompeakqualtools package, RRID:SCR_005331). The resulting NSC (=1.24) and RCS (=0.82) values met the ENCODE quality thresholds (NSC $\leq$ 1.05, RCS $\leq$ 0.8) (*Landt et al., 2012*). Peaks of enriched Satb2 binding were called using MACS2 (RRID:SCR_013291) (*Zhang et al., 2008*) (version 2.1.0), by allowing only one tag at the same location and setting the false discovery rate to 0.01 and the fold-enrichment cutoff to 2. This resulted in 8414 high confidence peaks and a fraction of reads in peaks (FriP) of 6.2% which is well above the ENCODE requirements for good quality data. Peak annotation and comparisons with other public data sets were performed with the R/Bioconductor packages ChIPseeker (*Yu et al., 2015*) and ngs.plot (RRID:SCR_011795) (*Shen et al., 2014*). Testing of ChIP-seq peak data for the enrichment of biological pathways and Gene Ontology terms was performed by using ChIP-Enrich tool (*Welch et al., 2014*). Public datasets (GSE63271, GSE66701, GSE21161, GSE65159) (*Gjoneska et al., 2015*; *Kim et al., 2010*; *Telese et al., 2015*; *Wang et al., 2015*) were downloaded from GEO and genomic coordinates were lifted over to mouse mm10 for data sets which were only available in a differing genome assembly version. Datasets that did not contain peak data (BED files) were reanalyzed by mapping quality filtered raw reads to mm10 and calling peaks using MACS2.

## ChIP

ChIP was performed on microdissected adult mouse CA1 tissue (a pool of 8–10 mice) by using the Zymo-Spin ChIP kit (Zymo Research) following manufacturer's instructions. Briefly, 100 mg of CA1 tissue were cross-linked with 1% formaldehyde (Sigma) for 10 min at RT and neutralized with 0.125M glycine. Chromatin was fragmented using a Bioruptor Plus sonicator (Diagenode Inc., Denville, NJ), yielding fragment size range of 200–500 bp. Sonicated chromatin (10–15 µg) was incubated at 4°C overnight with 5 µg antibodies (ab34735, AB_2301417, Abcam or normal rabbit IgG 12–370, AB_145841, Millipore). After washing, ChIP-ed DNA was eluted; reverse cross-linked at 65°C for 2 hr and purified using Zymo-Spin ChIP kit (Zymo Research).

## Behavior experiments

### Contextual fear conditioning

Contextual fear conditioning was performed in a 25 × 25 × 35 cm chamber with transparent walls and a metal rod floor, cleaned with water and illuminated to 300 lux (TSE, Bad Homburg, Germany) as previously described (**Busquet et al., 2008**; **Dobi et al., 2013**; **Sartori et al., 2011**). After a 120 s acclimation period, mice were conditioned with three presentations of a 0.60 mA scrambled foot shock, with a 120 s inter-shock interval. The mice were allowed to remain in the chamber for an additional 120 s following the last stimulus presentation. Short-term and long-term fear memories were tested 1 hr and 24 hr later respectively in the conditioning chamber. Freezing was measured as an index of fear (**Blanchard and Blanchard, 1969**) manually scored based on DVD recordings, defined as no visible movement except that required for respiration, and converted to a percentage [(duration of freezing /total time) × 100] by a trained observer blind to the animals' group/genotype.

### Object location and novel object recognition paradigms

Prior to training, mice were handled 1–2 min for five days and then habituated to the experimental apparatus (a 41 × 41 × 41 cm open field arena containing home-cage floor bedding and illuminated to 150 Lux; Tru Scan, Coulbourn Instruments, Holliston, MA) for 5 min a day for three consecutive days in the absence of objects. During the training period, mice were placed into the experimental apparatus containing two identical objects (blue colored Lego Duplo blocks 2.5 × 2.5 × 5 cm) and allowed to explore for 10 min. During the short-term (1 hr) or long-term (24 hr) retention tests, mice were placed in the experimental apparatus for 5 min. For assessment of spatial object location-dependent memory, one copy of the familiar object was placed in the same location as during the training trial, and one copy of the familiar object was moved and placed in the middle of the box. For the novel object recognition test (**Antunes and Biala, 2012**), one copy of the familiar object and a new object (100 ml glass beaker) were placed in the same location as during the training trial. Exploration was scored when the mouse's nose touched the object. All training and testing trials were videotaped and analyzed by individuals blind to the genotype of subjects. The relative exploration time (t) was recorded and expressed as a percent discrimination index (D.I. = $(t_{novel} - t_{familiar})$ / $(t_{novel} + t_{familiar})$ × 100%). Mean exploration times were then calculated and the discrimination indexes between treatment groups compared. Animals that explored less than 3 s total for both objects during either training or testing were removed from the analysis.

### Flinch-jump test

Reactivity to the foot shock was evaluated in the same apparatus used for contextual fear conditioning as previously described (**Sartori et al., 2011**). After a 120 s acclimation period, mice were subjected to a series of 1 s shocks of gradually increasing amperage (0.1 mA every 30 s) starting from 0.1 mA. Mice were scored for their first visible response to the shock (flinch), their first pronounced motor response (run or jump), and their first vocalized distress, as previously described (**Wittmann et al., 2009**).

## Slice electrophysiology

Acute hippocampal slices were prepared from 2–3 month old mice. In brief, mice were anesthetized and decapitated; the brain was quickly transferred into ice-cold carbogenated (95% $O_2$, 5% $CO_2$) artificial cerebrospinal fluid (ACSF) for 3 min. The ACSF used for electrophysiological recordings contained 125 mM NaCl, 2.5 mM KCl, 1.25 mM $NaH_2PO_4$, 2 mM $MgCl_2$, 26 mM $NaHCO_3$, 2 mM $CaCl_2$, and 25 mM glucose. Hippocampi were cut with a vibratome (VT 101200S; Leica, Germany) into 400 µm thick transversal slices. Recordings were performed in a submerged recording chamber at 32°C. Field excitatory postsynaptic potentials (fEPSPs) were recorded in the stratum radiatum of the CA1 region with a glass micropipette (resistance: 3–15 MΩ) filled with 3 M NaCl at a depth of ca. 150–200 µm. Monopolar tungsten electrodes were used for stimulating the Schaffer collaterals at a frequency of 0.1 Hz. Stimulation was set to elicit a fEPSP with a slope of ca. 40–50% of maximum for LTP recordings. After 40 min of baseline stimulation, LTP was induced by applying theta-burst stimulation (TBS), in which a burst consisted of 4 pulses at 100 Hz which were repeated 10 times in a 200 ms interval (5 Hz). Three of such trains were used to induce LTP at 0.1 Hz. Basic synaptic transmission and presynaptic properties were analyzed via input-output (IO) measurements and paired-pulse

facilitation (PPF). The IO-measurements were performed by application of a defined value of current (25–250 μA in steps of 25 μA). PPF was performed by applying a pair of two stimuli by different inter-stimulus intervals (ISI) ranging from 10, 20, 40, 80 to 160 ms. Data were collected, stored, and analyzed with LABVIEW software (National Instruments, Austin, TX). The initial slope of fEPSPs elicited by stimulation of the Schaffer collaterals was measured over time, normalized to baseline, and plotted as average ± SEM. Statistical analyses were performed using Student's t-test.

### Statistical analysis

Statistical analysis was conducted as indicated in the figure legends using SPSS software (SPSS Inc). Statistical significance was determined by Student's t-test, one-way ANOVA or two-way ANOVA followed by appropriate *post hoc* test (Tukey, Hochberg, Bonferroni or Fisher's LSD). Data represent mean ± SEM of at least three independent biological replicates. No statistical methods were used to predetermine sample sizes; however our sample sizes were similar to those reported in previous studies. The data distribution was assumed to be normal, but it was not formally tested, except for datasets with $n > 10$.

### Accession codes

The RNA-seq and ChIP-seq data sets are deposited at GEO under accession number GSE77005.

## Acknowledgements

We are grateful to P Feurle for excellent technical assistance. We acknowledge N Yannoutsos for ES cell injection. We are grateful to F Ferraguti for support in stereotaxic injections and critical reading of the manuscript. This work was funded by the Austrian Science Fund (FWF grants DK W1206 'Signal Processing in Neurons' to GD and NS; SFB F44 'Cell Signaling in Chronic CNS Disorders' to GA, GD and NS, P25014-B24 to GA), Medical University of Innsbruck (MUI-Start 2010012004 to GA), and the Deutsche Forschungsgemeinschaft (to MK).

## Additional information

#### Funding

| Funder | Grant reference number | Author |
| --- | --- | --- |
| Deutsche Forschungsgemeinschaft | | Martin Korte |
| Austrian Science Fund | Graduate Programme 'Signal Processing in Neurons' (SPIN) W 1206 | Nicolas Singewald Georg Dechant |
| Austrian Science Fund | Special Research Program (SFB) F44 (F4410-B23) 'Cell Signaling in Chronic CNS Disorders' | Nicolas Singewald |
| Austrian Science Fund | Special Research Program (SFB) F44 (F4416-B23) 'Cell Signaling in Chronic CNS Disorders' | Georg Dechant Galina Apostolova |
| Austrian Science Fund | P25014-B24 | Galina Apostolova |
| Innsbruck Medical University | MUI-Start 2010012004 | Galina Apostolova |

The funders had no role in study design, data collection and interpretation, or the decision to submit the work for publication.

#### Author contributions

CJ, CR, AA, AD, GJ, IC, Final aproval of the version to be published, Acquisition of data, Analysis and interpretation of data; NW, Final aproval of the version to be published, Conception and design, Acquisition of data, Analysis and interpretation of data; DR, Final aproval of the version to be published, Analysis and interpretation of data, Drafting or revising the article; MK, Final aproval of the

version to be published, Conception and design, Analysis and interpretation of data; AF, FS, NS, GD, Final aproval of the version to be published, Conception and design, Analysis and interpretation of data, Drafting or revising the article; GA, Conception and design, Acquisition of data, Analysis and interpretation of data, Drafting or revising the article

### Author ORCIDs

Andrea Delekate, http://orcid.org/0000-0002-8887-0806
Galina Apostolova, http://orcid.org/0000-0003-2682-4385

### Ethics

Animal experimentation: All animal experimentation procedures were approved by the Austrian Animal Experimentation Ethics Board (Permit Number: GZ: BMWFW-66.011/0078-WF/II/3b/2014)

## Additional files

### Supplementary files

• Supplementary file 1. List of primers used in RT-qPCR and ChIP-qPCR analyses.

### Major datasets

The following dataset was generated:

| Author(s) | Year | Dataset title | Dataset URL | Database, license, and accessibility information |
|---|---|---|---|---|
| Apostolova G, Sananbenesi F, Dechant G, Rieder D | 2016 | The Schizophrenia Risk Gene Product Satb2 Regulates miRNAs Expression and Long-Term Memory in Adult CNS | https://www.ncbi.nlm.nih.gov/geo/query/acc.cgi?acc=GSE77005 | Publicly available at the NCBI Gene Expression Omnibus (accession no: GSE77005) |

The following previously published datasets were used:

| Author(s) | Year | Dataset title | Dataset URL | Database, license, and accessibility information |
|---|---|---|---|---|
| Rosenfeld MG, Wang J | 2014 | A Novel Neuron-specific Histone H4K20 Demethylase LSD1n Promotes Transcriptional Elongation and is Essential for Learning and Memory | https://www.ncbi.nlm.nih.gov/geo/query/acc.cgi?acc=GSE63271 | Publicly available at the NCBI Gene Expression Omnibus (accession no: GSE63271) |
| Telese F, Ma Q | 2015 | LRP8-Reelin-regulated Neuronal (LRN) Enhancer signature underlying learning and memory formation (ChIP-Seq) | https://www.ncbi.nlm.nih.gov/geo/query/acc.cgi?acc=GSE66701 | Publicly available at the NCBI Gene Expression Omnibus (accession no: GSE66701) |
| Kim T, Hemberg M, Gray JM, Kreiman G, Greenberg ME | 2010 | Widespread transcription at neuronal activity-regulated enhancers | https://www.ncbi.nlm.nih.gov/geo/query/acc.cgi?acc=GSE21161 | Publicly available at the NCBI Gene Expression Omnibus (accession no: GSE21161) |
| Gjoneska E, Pfenning AR, Kundaje A, Tsai L, Kellis M | 2015 | Conserved epigenomic signatures between mouse and human elucidate immune basis of Alzheimer's disease | https://www.ncbi.nlm.nih.gov/geo/query/acc.cgi?acc=GSE65159 | Publicly available at the NCBI Gene Expression Omnibus (accession no: GSE65159) |

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
