## [Decision Letter]

Thank you for submitting your article "Satb2 determines miRNA Expression and Long-Term Memory in Adult CNS" for consideration by *eLife*. Your article has been favorably evaluated by a Senior Editor and three reviewers, one of whom is a member of our Board of Reviewing Editors. The reviewers have opted to remain anonymous.

The reviewers have discussed the reviews with one another and the Reviewing Editor has drafted this decision to help you prepare a revised submission.

Summary:

The chromatin-associated protein Satb2 is known to play important roles in neuronal fate determination in the developing brain, however its functions in mature neurons have not been explored. In this manuscript the authors use a *Camk2-Cre* to conditionally knockout Satb2 in mature forebrain neurons revealing impairments in hippocampal synapse plasticity and memory consolidation. in vivo RNAseq experiments revealed that a few dozen mRNAs and several hundred miRNAs were differentially expressed in hippocampus of cKO mice. They choose Arc as a potential microRNA target, found that its protein expression was reduced in Satb2 cKO hippocampus, and saw improved memory performance when Arc expression was rescued. Intermingled with these experiments, in cultured neurons they found that expression of Satb2 was induced by exogenous BDNF or bicuculline and showed that overexpressed Satb2 binds the promoters of both protein coding and non-coding genes, including a number of microRNA genes.

All three reviewers were convinced of the high quality of the data and the novelty of the behavioral findings. However, all three raised concerns about the links between the mechanistic data and the behavioral data. Some of these concerns can be addressed in the text or through additional data analysis, but a few new experiments are essential to make the mechanistic data on Satb2 chromatin binding a compelling story to explain the behavioral results.

Essential revisions:

1) A major weakness of the ChIP experiments is that they are performed in cultured neurons with overexpressed Satb2 whereas the RNAseq data are derived from hippocampus of the cKO mice. To confirm that the culture/overexpression ChIP experiments are relevant for the in vivo gene expression data the authors need to confirm by ChIP-PCR that endogenous Satb2 is bound to these same regions in hippocampus of the cKO mice. Furthermore, as a control for the specificity of the V5 antibody in the overexpression ChIP, the minimum control is ChIP from uninfected neurons – this could be ChIP-PCR as well.

2) Several aspects of the analysis and description of the mRNAs and miRNA expression data need to be improved.

The authors overemphasize the miRNA findings at the expense of discussing the mRNAs that are changed. In principle, there is nothing from the ChIP data selectively linking Satb2 with miRNA expression that justifies the emphasis on miRNAs when interpreting the behavioral results.

The miRNA RNAseq data need to be validated by a second method just like the mRNA data are validated.

A number of pre-miRNAs are usually detected in mRNAseq experiments, but apparently they are not differentially expressed in cKO mice. Could this suggest that Satb2 does not alter miRNA levels by regulating their transcription?

The authors report that nearly 50% of all microRNAs are dysregulated in Satb2 cKO mice, whereas only a very small number of mRNAs show differences. This is not clearly explained by the ChIP data (which show Satb2-V5 bound to both sets of promoters). Some explanation should be offered.

3) As presented, the BDNF-dependent regulation of Satb2 doesn't help to build a cogent model to explain the functions of Satb2 in learning and memory or LTP. Do the authors think that regulation of expression contributes to the requirement for Satb2 in learning and memory? If so it would be interesting to see if activity-dependent expression of Arc is dysregulated in Satb2 knockouts. Do physiologically relevant stimuli induce the expression of Satb2 in vivo? If these data cannot be more tightly tied to the behavior they would be better off moved to the supplementary data.

[Editors' note: further revisions were requested prior to acceptance, as described below.]

Thank you for resubmitting your work entitled "Satb2 determines miRNA expression and long‐term memory in adult central nervous system" for further consideration at *eLife*. Your revised article has been favorably evaluated by a Senior Editor, and a Reviewing Editor.

The manuscript has been improved but there are some remaining issues that need to be addressed before acceptance, as outlined below:

In this revision the authors have provided essential additional data needed to support the findings of this study. In particular, the chromatin immunoprecipitation controls (e.g. the V5 ChIP from nontransduced neurons and the endogenous Satb2 ChIP from hippocampus) provide crucial support for the model of direct transcriptional regulation by Satb2 in hippocampal neurons of the mRNA and miR targets identified as dysregulated in the sequencing studies. Further the validation of the mIR findings by RT-PCR and ChIP-PCR strengthen the focus of the manuscript on these targets.

The major finding of this study – that Satb2 has an important gene regulatory function in mature hippocampal neurons relevant to synaptic plasticity and learning and memory – is a novel and useful addition to the literature that will be of interest to a broad range of neurobiologists.

That being said, the narrative of the manuscript still overstates the impact of some of the findings and requires further textual revision to fully address the original critiques from the first round of review.

1) The BDNF data remain partially disconnected from the rest of the story and the "synapse to nucleus feedback loop" idea that is repeated several times in the manuscript is not supported by the data. The experiments demonstrating BDNF-dependent regulation of Satb2 expression in cultured neurons are well done, and the authors make a reasonable argument to keep these data in the main figures. However, they do not have relevant evidence for BDNF-dependent regulation of Satb2 in vivo or any evidence that induction of Satb2 expression is required for its role in hippocampal function. (Dark rearing is not a relevant experiment to demonstrate activity- or BDNF-dependent gene regulation in vivo. The more usual experiment would be dark adaption after eye opening followed by light exposure, since dark rearing induces developmental delays.) I do not think it is required that the authors demonstrate BDNF-dependent regulation of Satb2 expression in vivo. However in lieu of these data or other data showing that BDNF- or activity-dependent regulation of Satb2 levels matter for the functions of Satb2 identified in this study, then the authors cannot conclude in the Discussion that they have identified a synapse to nucleus feedback circuit. This language needs to be removed from the last line of the Abstract, the last line of the Introduction, and the first line of the Discussion. It is reasonable based on the BDNF data provided that the authors can speculate in the Discussion that BDNF-dependent regulation of Satb2 might regulate synapse plasticity as they propose. Finally, the Discussion section on the relationship between BDNF and Satb2 in synapse plasticity (third paragraph) remains beyond the data. Lots of things disrupt LTP in the hippocampus in addition to BDNF and Satb2, so just because those two have LTP disruption in common does not bind them tightly together.

2) The authors have improved their commentary on the possibility that Satb2 is acting to regulate mature hippocampal neuronal function via chromatin looping, but several sentences in the Discussion still remain that are overstatements from the data presented.

"Our findings suggest that some of the key functions of the hippocampus depend on changes in the higher-order chromatin architecture." The authors' findings suggest these functions "may" depend on chromatin looping. Yes, the literature shows that Satb2 regulates chromatin architecture but it very well is likely to have additional functions. Unless the authors were to study chromatin architecture in these neurons of this mouse, it cannot be concluded that is the mechanism of action in this study.

"Our results in primary hippocampal cultures also suggest that this type of higher-order chromatin rearrangement is an activity- and BDNF dependent process that involves changes in Satb2 expression levels." This is speculation since this manuscript provides no evidence for changes in chromatin looping associated with changes in BDNF-induced Satb2 expression. This can be proposed, but the data do not "suggest" this conclusion.

"Our findings indicate Satb2 as a regulator of such 3D-chromatin configurations at a genome-wide level. " There is no data at all in this manuscript that is relevant to this statement. It should be removed.

3) Finally, there are multiple lines in the text where vague words are used to describe the importance of the findings. These should be corrected to be more quantitative and accurate. Examples are below:

Subsection “BDNF and synaptic activity up-regulate Satb2 via the ERK1/2 pathway”, second paragraph: The two fold induction of mRNA expression is statistically significant but not "strong".

Subsection “Satb2 occupies active gene promoters”, first paragraph: "suggesting that Satb2 predominantly binds to active promoters of transcription factors and synaptic plasticity genes." It is not clear the authors have the data to support the word "predominantly" which would be quantified as well more than half of the targets are these genes. Unless this fact can be provided the phrase should be cut from the sentence.

Subsection “Satb2 occupies active gene promoters”, third paragraph: "our analysis revealed strong association of Satb2 with mIR promoters" The word "strong" is not quantitative. Do the authors have a statistical way to show this is "significant"? Otherwise the observation should just be stated with no qualifier.

Subsection “Satb2 determines the expression of protein-coding genes and miRNAs linked to learning and memory in the CA1 hippocampal field”, last paragraph: The word "pivotal" for the role of Satb2 in microRNA regulation is very strong and would suggest it is more important than other mIR regulators. "Important" would be more accurate for the data presented here.

Subsection “Re-expression of Satb2 into adult hippocampus rescues both a decrease in Arc protein and fear memory deficits”, first paragraph: "A large group" of microRNAs regulating Arc is a vague term that should be quantified or restated.

Discussion, fifth paragraph: The preferential regulation of miRNAs over mRNAs does not "indicate" the presence of Satb2 cofactors that determine selective binding and control of miR promoters – that is just one possible explanation.

---

## [Author Response]

*Essential revisions:*

*1) A major weakness of the ChIP experiments is that they are performed in cultured neurons with overexpressed Satb2 whereas the RNAseq data are derived from hippocampus of the cKO mice. To confirm that the culture/overexpression ChIP experiments are relevant for the* in vivo *gene expression data the authors need to confirm by ChIP-PCR that endogenous Satb2 is bound to these same regions in hippocampus of the cKO mice. Furthermore, as a control for the specificity of the V5 antibody in the overexpression ChIP, the minimum control is ChIP from uninfected neurons – this could be ChIP-PCR as well.*

ChIP-qPCR using chromatin from CA1 hippocampal tissue and an antibody against endogenous Satb2 is a highly demanding, and technically challenging experiment for the following reasons:

A) There is no commercially available ChIP-grade antibody against Satb2.

B) Our own unpublished data show that Satb2 levels in the adult CA1 hippocampal area are approximately 10-20-fold lower compared to the levels in embryonic or neonatal cortex, where the previously published ChIP assays have been carried out.

Despite these substantial technical hurdles we were able to confirm Satb2 binding sites initially revealed by in vitro ChIP-seq experiments in CA1 tissue. In these newly added experiments we used a rabbit polyclonal antibody against Satb2 (offered by Abcam, ad34735) that has been previously used in EMSA assays and in a single study describing ChIPseq of chromatin derived from embryonic cortex (Mutual regulation between *Satb2* and *Fezf2* promotes subcerebral projection neuron identity in the developing cerebral cortex, McKenna et al., 2015). As a negative control, we used chromatin from Satb2 cKO CA1 tissue. With this Satb2-specific antibody we succeeded in demonstrating Satb2 enrichment at various Satb2 target regions that we had previously identified in our in vitro ChIPseq in chromatin ex vivo samples from control but not Satb2 cKO mice. The results are included in the revised version of the manuscript (Figure 5—figure supplement 4).

We also carried out ChIP qPCR assays using AAV-Satb2-V5 transduced and non-transduced primary hippocampal neurons. We performed one ChIP assay using AAV-Satb2-V5 transduced primary neurons (50 million cells) and at least two independent ChIP assays using non- transduced cultures (50 million cells per assay). The experiments in non-transduced cells showed minimal interaction with Satb2 for all tested target regions. The results are included in the revised version (Figure 5—figure supplement 1). Furthermore, we complemented the ChIP qPCR data derived from AAV-Satb2-V5 transduced neurons with a second biological replicate of the ChIP seq experiment (in which libraries for input and pooled ChIP DNA from three independent ChIP assays were used). The results from the second biological replicate completely confirmed the mapped Satb2 genomic binding sites and will be deposited at the GEO.

*2) Several aspects of the analysis and description of the mRNAs and miRNA expression data need to be improved.*

*The authors overemphasize the miRNA findings at the expense of discussing the mRNAs that are changed. In principle, there is nothing from the ChIP data selectively linking Satb2 with miRNA expression that justifies the emphasis on miRNAs when interpreting the behavioral results.*

We agree with the reviewers about this point. We have changed the Results and Discussion sections accordingly.

*The miRNA RNAseq data need to be validated by a second method just like the mRNA data are validated.*

New experimental data have been added as required: Figure 6—figure supplement 2 Validation of the differential expression of selected miRNAs by qPCR.

*A number of pre-miRNAs are usually detected in mRNAseq experiments, but apparently they are not differentially expressed in cKO mice. Could this suggest that Satb2 does not alter miRNA levels by regulating their transcription?*

Pre-miRNAs were not detected in our RNAseq and sRNAseq experiments. We used the TruSeq RNA v2 kit which relies on polyA selection and on essence precludes the analysis of pre-miRNAs. The sRNAseq protocol we applied also does not allow pre-miRNAs quantification because of size selection during library prep (only fragments with 147-157 bp length including adaptors which correspond to miRNAs (22 bp) and piwi-interacting RNAs or some other regulatory small RNA molecules (about 35 bp) are selected).

*The authors report that nearly 50% of all microRNAs are dysregulated in Satb2 cKO mice, whereas only a very small number of mRNAs show differences. This is not clearly explained by the ChIP data (which show Satb2-V5 bound to both sets of promoters). Some explanation should be offered.*

A potential explanation is added in the Discussion.

*3) As presented, the BDNF-dependent regulation of Satb2 doesn't help to build a cogent model to explain the functions of Satb2 in learning and memory or LTP. Do the authors think that regulation of expression contributes to the requirement for Satb2 in learning and memory? If so it would be interesting to see if activity-dependent expression of Arc is dysregulated in Satb2 knockouts. Do physiologically relevant stimuli induce the expression of Satb2* in vivo*? If these data cannot be more tightly tied to the behavior they would be better off moved to the supplementary data.*

Our data in primary cultures reveal the potential of the Satb2 promoter to respond to BDNF-trkB signaling in hippocampal neurons. In our opinion, this is a novel finding deserving description in the main text since to our knowledge a similar mode of regulation has not so far been demonstrated for any other transcription factor required for learning and memory apart from immediate early genes. These data give us grounds to hypothesize that Satb2 is subjected to the same type of regulation also in vivo, e.g. in CA1 following behavioral learning or in the visual cortex after modulation of sensory input. Our behavior results showing that some of the best investigated functions of BDNF in the hippocampus, i.e. modulation of late-LTP and memory formation, depend on Satb2 are consistent with this hypothesis. However, it is very challenging to reveal regulation of Satb2 in vivo in the CA1 hippocampal area, following associative learning for the following reasons:

The use of whole CA1 hippocampal homogenates (e.g. in Western blotting experiments) will dilute the signal when only a small proportion of the cells in the sample are responsible for the changes. Tagging of activated neurons, e.g. in H2B-GFP TetTag mice, has shown that only 10-15% of CA1 pyramidal neurons become activated and labeled during fear conditioning (Reactivation of Neural Ensembles during the Retrieval of Recent and Remote Memory, Taylor et al., 2013). Thus, a special transgenic line should be used (such as H2B-GFP TetTag mice) to permanently tag neurons that are active during contextual fear conditioning and then examine Satb2 levels exclusively in the labeled neurons.

To circumvent this issue we used the dark-rearing paradigm to manipulate activity in the visual cortex. Unlike learning paradigms where only a small proportion of neurons are engaged, visual deprivation upon dark rearing causes a more general and robust decrease in activity levels in the visual cortex making it a more suitable model. Dark-rearing of mice for 4 days lead to a significant decrease in Egr1 levels in all layers of visual cortex (Figure 8) as previously shown (Differential induction and decay curves of c-fos and zif268 revealed through dual activity maps; Zangenehpour and Chaudhuri, 2002). Similarly, we found a small, but significant reduction in Satb2 levels by dark-rearing (Figure 8). This finding demonstrates the possibility of Satb2 regulation by neuronal activity. The small effect size that we observed points to the necessity of more specific measurements in learning paradigms such as determining Satb2 in activity-tagged cells or after mosaic DREADD silencing. In addition, a circumstantial evidence for Satb2 regulating or being regulated by ensemble dynamics is an observation of variability of Satb2 nuclear intensities between cells. It is currently under investigation. We did not include these data in the current manuscript because we consider them beyond the scope of the current investigations focusing on Satb2 function in hippocampus.

Author response image 1.(**A**) Mean intensity of Egr1 in each layer was normalized to corresponding layer 4 for each mouse to obtain relative Egr1 expression. Dark-rearing leads to a significant reduction in relative Egr1 levels in layer 2/3 and 6. (**B**) Dark-rearing leads to a reduction in relative Satb2 levels in layer 2/3 and 6.**DOI:**
http://dx.doi.org/10.7554/eLife.17361.028

We also tested the idea of Satb2 regulation being necessary for learning-induced Arc expression by using IHC with tyramide signal amplification after contextual fear conditioning. The experiment compared three groups, naïve animals, 2h and 12h after training for both genotypes with 6 mice per group. Arc intensity was measured in a region of interest within the dorsal CA1-stratum pyramidale. The analysis did not reveal any effect of Satb2 loss on either early or late wave of Arc induction. Here, we would like to point out that intensity measurements by IHC cannot be directly compared to Western blot quantification of Arc protein in the CA1 area since in IHC experiment Arc expression in the cell soma is only considered whereas Western blot measures also Arc in dendrites where Arc mRNA is translated.

[Editors' note: further revisions were requested prior to acceptance, as described below.]

*The manuscript has been improved but there are some remaining issues that need to be addressed before acceptance, as outlined below:*

*In this revision the authors have provided essential additional data needed to support the findings of this study. In particular, the chromatin immunoprecipitation controls (e.g. the V5 ChIP from nontransduced neurons and the endogenous Satb2 ChIP from hippocampus) provide crucial support for the model of direct transcriptional regulation by Satb2 in hippocampal neurons of the mRNA and miR targets identified as dysregulated in the sequencing studies. Further the validation of the mIR findings by RT-PCR and ChIP-PCR strengthen the focus of the manuscript on these targets.*

*The major finding of this study – that Satb2 has an important gene regulatory function in mature hippocampal neurons relevant to synaptic plasticity and learning and memory – is a novel and useful addition to the literature that will be of interest to a broad range of neurobiologists.*

*That being said, the narrative of the manuscript still overstates the impact of some of the findings and requires further textual revision to fully address the original critiques from the first round of review.*

*1) The BDNF data remain partially disconnected from the rest of the story and the "synapse to nucleus feedback loop" idea that is repeated several times in the manuscript is not supported by the data. The experiments demonstrating BDNF-dependent regulation of Satb2 expression in cultured neurons are well done, and the authors make a reasonable argument to keep these data in the main figures. However, they do not have relevant evidence for BDNF-dependent regulation of Satb2* in vivo *or any evidence that induction of Satb2 expression is required for its role in hippocampal function. (Dark rearing is not a relevant experiment to demonstrate activity- or BDNF-dependent gene regulation* in vivo*. The more usual experiment would be dark adaption after eye opening followed by light exposure, since dark rearing induces developmental delays.) I do not think it is required that the authors demonstrate BDNF-dependent regulation of Satb2 expression* in vivo*. However in lieu of these data or other data showing that BDNF- or activity-dependent regulation of Satb2 levels matter for the functions of Satb2 identified in this study, then the authors cannot conclude in the Discussion that they have identified a synapse to nucleus feedback circuit. This language needs to be removed from the last line of the Abstract, the last line of the Introduction, and the first line of the Discussion. It is reasonable based on the BDNF data provided that the authors can speculate in the Discussion that BDNF-dependent regulation of Satb2 might regulate synapse plasticity as they propose. Finally, the Discussion section on the relationship between BDNF and Satb2 in synapse plasticity (third paragraph) remains beyond the data. Lots of things disrupt LTP in the hippocampus in addition to BDNF and Satb2, so just because those two have LTP disruption in common does not bind them tightly together.*

*2) The authors have improved their commentary on the possibility that Satb2 is acting to regulate mature hippocampal neuronal function via chromatin looping, but several sentences in the Discussion still remain that are overstatements from the data presented.*

We have eliminated all reference to a “synapse to nucleus feedback loop” in the Abstract, Introduction and Discussion sections.

The discussion of functional interactions between BDNF and Satb2 has been slashed. Although we are convinced of the proposed BDNF and activity-dependent regulation of Satb2 in vivo, we agree that the in vitro data in the present manuscript, although providing strong indications, have been over-interpreted by us. Work concerning Satb2 regulation by dark adaptation/re-exposure to light in V1 (which in fact was what we were referring to in our previous letter) is in progress in our laboratory and we will discuss mechanism of in vivo-Satb2 regulation and its functional connex with BDNF in due time.

*3) Finally, there are multiple lines in the text where vague words are used to describe the importance of the findings. These should be corrected to be more quantitative and accurate. […]*

In addition to eliminating some grammatical errors, we have erased or changed a number of vague terms and expletives throughout the manuscript.